

# The Lagrangian moisture source and transport diagnostic WaterSip V3.2

Harald Sodemann[1,2]

[1]Geophysical Institute, University of Bergen, Norway
[2]Bjerknes Centre for Climate Research, Bergen, Norway

**Correspondence:** Harald Sodemann (harald.sodemann@uib.no)

**Abstract.** WaterSip is a diagnostic software tool that identifies the evaporation sources and transport pathways of precipitation or water vapour over a target area based on Lagrangian model output. In addition to the geographic location, WaterSip identifies select thermodynamic properties of the moisture sources, during atmospheric transport, and during arrival over the target area. WaterSip software thereby employs the Lagrangian diagnostic algorithm for quantitative moisture source accounting of Sodemann et al. (2008b). The software tool requires output from Lagrangian particle dispersion models or trajectory models as input for the diagnostic. Moisture sources are then identified from changes in specific humidity along these trajectories at each output time step. The ratio between changes in specific humidity and the specific humidity of the air parcel allow to estimate the quantitative contribution of a moisture source to the air parcel at a specific time and location. Together with the temporal sequence, this provides the basis for identifying moisture source contributions to the final precipitation. WaterSip also identifies and aggregates further thermodynamic and geographic properties of the moisture source and during the moisture transport. Designed to operate on large datasets of regional to global domain-filling trajectories, WaterSip provides the results of the moisture source identification as gridded information in a variety of output files in netCDF format. This paper describes the relevant methodological foundations, the technical set-up and configuration, and provides a consistent example case study to illustrate the use and interpretation of the software tool and its results. Importantly, key uncertainties and caveats are described and discussed throughout the text. Users of WaterSip should be aware of these uncertainties to obtain a valid and reliable interpretation of the diagnostic results.

## 1 Introduction

Atmospheric dynamics transports water vapour from an evaporation source to a precipitation sink, manifested in the global spectrum of weather systems. Knowledge about how weather systems modulate the atmospheric water cycle from source to sink is key for understanding the causes of hydro-meteorological extremes (e.g., Winschall et al., 2014b), precipitation seasonality (e.g., Sodemann and Zubler, 2010), and climatological means (e.g., Brubaker et al., 1993). Process studies have looked at how oceans contribute to precipitation on land Stohl and James (2004), and how land areas recycle precipitation (Eltahir and Bras, 1996). On paleoclimate time scales, knowledge of the variation of sources and transport pathways of water





vapour are central to the interpretation of archives of the past hydroclimate, including ice cores (e.g., Johnsen et al., 1989; Sodemann et al., 2008a) and cave deposits at lower latitudes (e.g., Baker et al., 2015).

Measurements of precipitation and atmospheric water vapour do not directly reveal information about their evaporation sources. In specific situations may the stable water isotope composition of atmospheric waters allow to separate the contribution of source regions and phase changes during transport (Sodemann et al., 2008a; Aemisegger et al., 2014). In general, though, due to the multitude of influences, the problem is underdetermined, an then also water isotope measurements require the support of diagnostic numerical tools to disentangle the contribution of different processes and evaporation sources. As a consequence, results from moisture source studies often suffer from a lack of definitive, observable constraints.

Different approaches from atmospheric models and model data have been used over time to access information about the sources and transport conditions of water vapour. Conceptual studies evaluated the water balance in domains over a given region, for example to determine precipitation recycling over the Continental U.S. (Eltahir and Bras, 1994) and the Amazon region (Salati et al., 1979). The opposite extreme of complexity are approaches where water vapour tracers are released within a numerical atmospheric model's water cycle (Numaguti, 1999; Yoshimura et al., 2004; Sodemann and Stohl, 2009). Hereby, water vapour is transported forward within a secondary water cycle of the numerical model for selected 'tagged' subsets of total evaporation. Such tagging approaches require a very substantial computational effort to obtain detailed representation of the sources, and over seasonal to inter-annual time scales. As an intermediate in terms of complexity and computational demand, Lagrangian methods have been employed to diagnose the water budget along air parcel trajectories (Dirmeyer and Brubaker, 1999; Stohl and James, 2004). Since trajectories can be calculated off-line from stored atmospheric fields, for example reanalysis data, such an approach is computationally more efficient than tagging studies. In addition, Lagrangian methods can more easily provide the spatial extent of moisture sources, and thus provide an intuitive connections between source and receptor areas.

This paper describes the concepts and use of the WaterSip[1] software. WaterSip consists of an implementation of the widely used Lagrangian moisture source and transport diagnostic of Sodemann et al. (2008b). Combined with programme routines to provide different gridded and aggregated information in output data files, WaterSip provides means to characterise the source regions and transport properties of water vapour and precipitation from Lagrangian air parcel trajectories. The algorithm of the WaterSip software includes a processing step that quantifies the contribution from a moisture source to the water vapour or precipitation in a target area. This so-called accounting step also provides an additional layer of information of the moisture sources. This can then be used to provide, for example, the sea surface temperature at the moisture source ((Sodemann et al., 2008b)b), the fraction of land versus ocean surface contribution (Sodemann and Zubler, 2010), the duration and transport distance from evaporation to precipitation (Laederach and Sodemann, 2016), as well as the extent of mixing and rainout during transport (Dütsch et al., 2018).

---

[1]The name WaterSip originates from the name of an internal research project at NILU, where the author developed large parts of the present code. The name can be understood as a metaphor, where increases of the water vapour mixing ratio between time steps could be imagined as sips of water taken by an air parcel.



Since the original publication of the method by Sodemann et al. (2008b), numerous technical and conceptual developments
have taken place, which are either fragmented across several publications, or not coherently described in the current literature.
The main objective of this work is to provide a consistent reference point for the use and interpretation of the method and
software, including its wide array of additional diagnostic output of moisture transport characteristics. Users will therefore no
longer need to re-implement the diagnostic algorithm behind WaterSip themselves, and can focus on the application of the tool

to different subject areas. A bibliography overview over studies that have used WaterSip shows the global to regional coverage
of previous work, and the variety of study objectives, from event-scale to climatological studies, process studies, sensitivity
studies, and the interpretation of measurements and paleoclimate records (Table 1).

    The WaterSip software in current version 3.2 presented here enables to flexibly configure the different parameters of the
diagnostic algorithm, and provides easy access to a multitude of additional moisture source and moisture transport charac-

teristics obtained from the algorithm. WaterSip currently works with two common models providing air parcel trajectories
(LAGRANTO, Sprenger and Wernli (2015) and FLEXPART, Stohl et al. (2005)), and can be extended to read in trajectory
files produced from other trajectory calculation tools. Implemented as a C++ code with parallelisation in openMP, and writing
output files in netCDF-CF format, the WaterSip code can be easily integrated in common scientific computing environments
and workflows.

The paper starts with a coherent description of the working principle of the algorithm and configuration of a run (Sec. 2),
followed by practicalities such as input data, running the software (Sec. 3), and an output file description (Sec. 3.3). A case study
is used as a consistent example throughout to get users started with the diagnostic output capabilities and the interpretation,
thereby providing initial guidance in applying WaterSip to new research topics. Readers that are new to the method may
chose to first take a look at the description of the results from the example case (Sec. 3.4 and Sec. 4) before reading through

the method description. Being an off-line diagnostic based on air parcel trajectories, the WaterSip algorithm makes several
simplifying assumptions that introduce uncertainty to the results. Causes of uncertainty are mentioned throughout the text and
discussed in Sec. 5. Users are advised to think carefully about the inherent limitation of the method to obtain valid and reliable
conclusions from applying WaterSip during their research efforts.

## 2   Method

Since the original publication of the WaterSip method in 2008, there has been a considerable growth in studies with this algo-
rithm (Table 1. There have also been new conceptual and technical developments that require an updated reference publication.
Recent developments include the use of WaterSip with FLEXPART model output (Sodemann and Stohl, 2009), assessing the
role of convection and assimilation increments Sodemann (2020); Fremme et al. (2023), the distinction of mixing and rain-
out events during transport Dütsch et al. (2018), the analysis of lifetime distribution of water vapour(Winschall et al., 2014b;

Laederach and Sodemann, 2016; Sodemann, 2020), the identification of water vapour in addition to precipitation, prediction
of stable isotope characteristics (Aemisegger et al., 2014), as well as technical developments regarding input and output files,
configuration and output quantities.





**Table 1.** Previous studies using the WaterSip method. Studies have been classified by their study region, time scale, and study topic.

| Region | Time scale | Study topic | References |
|---|---|---|---|
| Central Europe | Climatological | Precipitation seasonality | Sodemann and Zubler (2010) |
| | Inter-annual | Interpretation of measurements | Aemisegger et al. (2014) |
| North Atlantic | Inter-annual | Interpretation of measurements | Steen-Larsen et al. (2015) |
| | Event-based | Precipitation extreme | Weng et al. (2021) |
| | Inter-annual | Interpretation of measurements | Zannoni et al. (2022) |
| Mediterranean | Event composite | Precipitation extremes | Winschall et al. (2014b) |
| | Event-based | Method comparison | Winschall et al. (2014a) |
| Greenland | Climatological | Method description | Sodemann et al. (2008c) |
| | Climatological | Ice core interpretation | Sodemann et al. (2008a) |
| | Inter-annual | Interpretation of measurements | Bonne et al. (2014) |
| | Climatological | Ice core interpretation | Osman et al. (2021) |
| | Event-based | Atmospheric river precipitation | Bonne et al. (2015) |
| Antarctica | Climatological | Ice core interpretation | Wang et al. (2013) |
| | Climatological | Precipitation seasonality | Sodemann and Stohl (2009) |
| | Climatological | Ice core interpretation | Masson-Delmotte et al. (2011) |
| | Climatological | Ice core interpretation | Winkler et al. (2012) |
| | Climatological | Ice core interpretation | Buizert et al. (2018) |
| | Event-based | Atmospheric river precipitation | Terpstra et al. (2021) |
| East Asia | Climatological | Process study | Fremme and Sodemann (2019) |
| | Climatological | Model sensitivity | Fremme et al. (2023) |
| | Climatological | Cave deposit interpretation | Baker et al. (2015) |
| India and Pakistan | Event | Precipitation extreme | Martius et al. (2012) |
| | Event composite | Precipitation extremes | Bohlinger et al. (2017) |
| | Event-based | Precipitation extreme | Bohlinger et al. (2018) |
| Global | Climatological | Precipitation life time | Laederach and Sodemann (2016) |
| | Climatological | Precipitation life time | Sodemann (2020) |
| | Climatological | Precipitation life time | Gimeno et al. (2021) |

## 2.1 Description of the WaterSip algorithm

Generally, the WaterSip algorithm works based on trajectories of atmospheric air parcels, described by a time-dependent
coordinate vector of longitude $\lambda$, latitude $\phi$, and height $z$:





$$\overrightarrow{x}(t) = [\lambda(t), \phi(t), z(t)]. \tag{1}$$

Trajectories on the way to a target area $A$ may pass over the underlying surface of either ocean (blue) or land regions (green), thereby acquiring moisture at the moisture sources (Fig. 1a). In order to determine the moisture sources of an air parcel, specific humidity $q$ and air temperature $T$, or relative humidity RH and $T$, must be available along the same trajectory, interpolated from the four-dimensional NWP model output fields $\boldsymbol{Q}$. Thus, $q$ becomes available along the transport pathway of air parcels towards a target area at every time step $i$:

$$q_i = \boldsymbol{Q}[\overrightarrow{x}(t = i \cdot \Delta t)]. \tag{2}$$

The conceptual starting point of the WaterSip algorithm is then the Lagrangian water budget, describing the change in specific humidity along an air parcel trajectory resulting from evaporation ($e$) and condensation ($c$) (Stohl and James, 2004):

$$\frac{Dq}{Dt} = e - p \tag{3}$$

The discretized Lagrangian water budget along a trajectory with time step $\Delta t$ of $\sim 1 - 6\,\mathrm{h}$ at time step $i$ (at time $t = i \cdot \Delta t$) is then

$$\frac{\Delta q_i}{\Delta t} = e_i - p_i. \tag{4}$$

Since the diagnostic looks backwards in time from the arrival of a trajectory at target area $A$, the arrival time is given by $t = 0$ at step 0, and $i$ increases backward along the trajectory (Fig. 1c). The change in specific humidity from the previous step, $i + 1$, to step $i$ is then obtained from

$$\Delta q_i = q_i - q_{i+1}. \tag{5}$$

Importantly, $\Delta q_i$ can either be positive, zero, or negative (Fig. **??**c). A positive $\Delta q$ (i.e., $\Delta q > 0$) is interpreted as a so-called moisture uptake, while a negative $\Delta q$ (i.e., $\Delta q < 0$) is interpreted as a moisture loss along the trajectory. Thereby, $\Delta q_i$ is always a net effect of both $e_i$ and $p_i$ that may have affected the water budget of the air parcel during time step $i$. The WaterSip algorithm necessarily assumes that either evaporation or precipitation exclusively act on an air parcel during discretisation time $\Delta t$, which introduces uncertainties depending on the discretisation time $\Delta$ (see Sec. 5). In practice, a threshold value is used to filter out small variations in $\Delta q$ that are due to numerical noise (see Sec. 2.5).

Each moisture uptake can contribute a share to $q_0$, the specific humidity at the last time step of a trajectory. These source contributions or uptakes are expressed by the fractional contribution $f$ of each of the $U$ moisture uptakes to the arriving water vapour:





$$q_0 = \sum_{n=1}^{U} f_n \cdot \Delta q_n + q_r, \qquad (6)$$

where $q_r$ is a residual amount of moisture that already is present in the air parcel at the earliest available time step of the trajectory (see Sec. 2.4.3). The objective of the WaterSip algorithm is now to obtain an estimate of the fractional contributions

of each moisture uptake $f_n$, which then allows to diagnose where, when, and under what conditions the air parcel water budget changed, and to what extent this is reflected in the precipitation falling from the air parcel.

At the time of a moisture uptake, the fractional contribution for any uptake $n$ is directly obtained from

$$f_n = \frac{\Delta q_n}{q_n}. \qquad (7)$$

However, if at a later time another moisture uptake contributes to the specific humidity in the same air parcel, $f_n$ needs to be

recalculated. This sequential updating of the fractional contributions is referred to as the *accounting* of specific humidity along the trajectory. Since Sodemann et al. (2008b) already provided a detailed example of this accounting procedure, only a short conceptual illustration is provided here.

In the case of another moisture uptake at a later time step $i$, the fractions $f_m$ of all previous uptakes $m > i$ simply need to be recalculated using Eq. 7 as

$$f'_m = \frac{\Delta q_m}{q_i} \text{for} m < i. \qquad (8)$$

In the case of a moisture loss (due to precipitation) at a later time $i$, some of the moisture contributed from previous uptakes $m < i$ will be lost. Since the WaterSip algorithm assumes that the water vapour is well mixed within one air parcel, precipitation events do not change the fractional contributions at that time. However, a moisture loss doe remove moisture from previous uptakes $m$ according to their fractional contribution at time $m$:

$$\Delta q'_m = \Delta q_m - f_m \cdot \Delta q_i. \qquad (9)$$

The accounting is complete when the above steps in Eq. (6-8) have been applied from the earliest time of the trajectory up to the arrival time $t = 0$. Dütsch et al. (2018) compactly expressed the final result of this accounting method in two update equations. While the algorithm is presented here in a detailed representation of each of the computational steps, the equations in Dütsch et al. (2018) are equivalent, and may facilitate a numerical implementation of the algorithm. WaterSip currently

implements the stepwise accounting procedure.

## 2.2 Precipitation estimate

Up to this point, the algorithm has provided quantitative information about the contributions of uptakes to the specific humidity in an air parcel at $\overrightarrow{x}(t = 0)$ at some height $z$. In order to transfer this information to surface precipitation at the arrival point





of trajectories, one needs to obtain an estimate of surface precipitation from these parcels. To this end, moisture loss from a
precipitating air parcel at the last time step before arrival ($\Delta q_0 < 0$) is assumed to directly contribute to surface precipitation.
To obtain a reliable estimate of the total precipitation in an area, a sufficiently large set of trajectories is required to faithfully
represent the atmospheric column above a target area (Fig. 1b). Then, the Lagrangian precipitation estimate $\tilde{P}$ (in units of kg
m$^{-2}$($\Delta t$)$^{-1}$) can be computed as the column integral of all $\Delta q_0^k$ for all $K$ trajectories residing over the target area $A$ at time
$t = 0$:

$$\tilde{P} = \frac{1}{A} \sum_{k=1}^{K} \left( \Delta q_0^k \cdot m^k \right), \tag{10}$$

Here, $m^k$ is the mass of air parcel $k$ residing over the target area. In case that trajectories are used rather than air parcels,
and that trajectories are started from a region that is discretised vertically in pressure intervals, Eq. 10 becomes

$$\tilde{P} = \frac{1}{g \cdot A} \sum_{k=1}^{K} \left( \Delta q_0^k \cdot \Delta p^k \right). \tag{11}$$

In practice, the contribution of each individual particle or trajectory is gridded onto the arrival grid using a gridding kernel
(see Sec. 4.3). According to the above definition of $\tilde{P}$, only air parcels with $\Delta q_0 < 0$ contribute to a precipitation event.
Therefore only air parcels fulfilling this requirement are considered in the analysis of precipitation moisture sources. To further
increase the reliability of the detection of precipitation events and computational efficiency, a minimum relative humidity
threshold during arrival is typically imposed in addition (see Sec. 2.5).

## 2.3 Accounted fraction

The sum of all fractional contributions from identified moisture uptakes along trajectory $k$, the total accounted fraction $f_{\text{tot}}^k$
provides information about the known sources of $q_0^k$ and thus $\Delta q_0^k$:

$$f_{\text{tot}}^k = \sum_{k=1}^{K} f_i^k. \tag{12}$$

Based on the vertical position (particle elevation) $z$ where the uptake is identified, the fractional contributions are subdivided
into the two categories: the accounted fraction in the boundary layer, $f_{\text{bl}}$, and the accounted fraction in the free troposphere,
$f_{\text{ft}}$. The sum of both fractions is the total explained fraction $f_{\text{tot}}$, which should obey the relation

$$f_{\text{tot}} = (f_{\text{bl}} + f_{\text{ft}}) < 1. \tag{13}$$

Then, $f_{\text{tot}}$ provides information about the consistency of the algorithm. In general, and on average, this relation is found to
be indeed fulfilled. However, if some threshold parameters are set outside a meaningful range of values (see Sec. 2.5), the total
explained fraction can exceed 1.0. Users are thus advised to inspect $f_{\text{tot}}$ in their results and to make adjustments if needed.





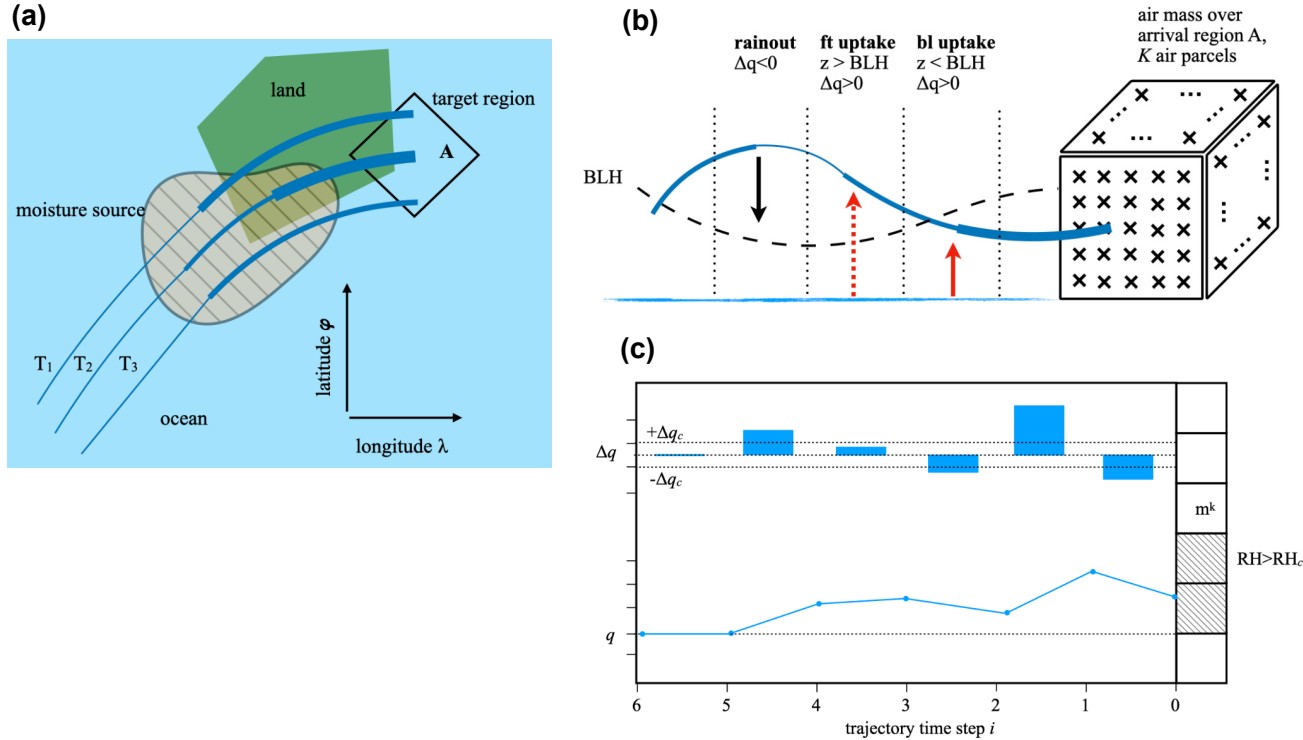

**Figure 1.** Schematic illustration of the moisture source diagnostic. (a) Horizontal view of a bundle of trajectories ($T_1, T_2, T_3$, dark blue lines) arriving in a target region A (black box). The trajectories pass over ocean (light blue) and land regions (green). An area of moisture uptake (dashed, orange) is detected where the specific humidity increases along trajectories (broadening blue lines). (b) 3-dimensional view of an air mass represented by air parcels that are traced backward using trajectories. Crosses denote arrival points in the air mass over the arrival domain A (black box). Locations of moisture increase from evaporation (red arrow) and decrease from precipitation (black arrow) are detected from changes in the specific humidity along trajectories (width of blue line). Moisture uptakes ($\Delta q > 0$) can be within the boundary layer (bl uptake) or in the free troposphere (ft uptake). Note that many trajectories at different vertical levels contribute to the Lagrangian precipitation estimate in the arrival region. Dashed black line denotes boundary-layer height (BLH). (c) Schematic of moisture uptakes and losses along 6 time steps of an air parcel with mass $m^k$ arriving with RH>RH$_c$ (shading). Note that some of the specific humidity changes are smaller than the critical threshold value $\Delta q_c$. See text for details.

The explained fractions can be used further to subdivide the Lagrangian precipitation estimate $\tilde{P}$ into its respective explained fractions. The total explained precipitation $\tilde{P}_{\text{tot}}$, for example, is obtained from

$$\tilde{P}_{\text{tot}} = \sum_{k=1}^{K} \left( f_{\text{tot}} \cdot \tilde{P}^k \right) = \frac{1}{A} \sum_{k=1}^{K} \left( f_{\text{tot}}^k \cdot \Delta q_0^k \cdot m^k \right). \tag{14}$$





Corresponding quantities $\tilde{P}_{\rm bl}$ and $\tilde{P}_{\rm ft}$ can be computed for $f_{\rm bl}$ and $f_{\rm ft}$. A choice to consider in this respect is which precipitation should be used for weighting when averaging diagnosed quantities, $\tilde{P}$ or only the explained fractions of ($\tilde{P}_{\rm tot}$, $\tilde{P}_{\rm bl}$,
$\tilde{P}_{\rm ft}$). Given that the Lagrangian precipitation estimate $\tilde{P}$ is typically in error by about 20–30% or more, this choice is probably secondary for the interpretation of the final results. In the WaterSip algorithm, $\tilde{P}$ is used, rather than $\tilde{P}_{\rm tot}$.

## 2.4  Trajectory input data

WaterSip uses Lagrangian particle trajectory data as its basic input. Trajectories that contain the vectors $\overrightarrow{x}^{k}$ with position and other necessary variables for the $K$ air parcels representing an air mass over the target region (Fig. 1b). Such trajectory data can
be obtained from different modelling tools. Currently, WaterSip is able to handle input from the Lagrangian trajectory model LAGRANTO (Sprenger and Wernli, 2015), and from the Lagrangian particle dispersion model FLEXPART (Stohl et al., 2005; Pisso et al., 2019). Minimum requirements with regards to the variables that are included on the trajectories are position, $T$, and $q$ or RH. The input data parameters are set in either settings group *Lagranto* (Table A7) or *Flexpart* (Table A6), depending on the used trajectory model. Here, the implications of discretisation choices of an air mass into air parcels are discussed, while
further technical details for creating input data are described in Sec. 3.

### 2.4.1  Particle mass $m_i$

An important user choice for the WaterSip diagnostic is the discretisation of the studied atmospheric mass into individual particles. In general, the more particles represent a particular air mass, the smaller the mass of air associated with one air parcel ($m^k$), and the more detailed and statistically reliable the results will become. For example, Sodemann and Stohl (2009) used
the global FLEXPART dataset of Stohl (2006) from a global domain-filling run with 2.4 million air parcels. With a total global atmospheric mass of $5.14 \times 10^{18}$ kg, each air parcel represents $2.14 \times 10^{12}$ kg of air. Laederach and Sodemann (2016) used 5 million air parcels, resulting in a particle mass of $m^k = 1.03 \times 10^{12}$ kg. Alternatively, the mass of an air parcel can also be specified in the RELEASES file of the FLEXPART model. For FLEXPART input data, the particle mass is set by parameter *partMass* (in kg) in the settings group *Flexpart* (Table A6).
If one uses LAGRANTO for the trajectory calculation, the atmospheric mass of one air parcel results from the calculation start grid. For LAGRANTO input data, the particle mass is first computed from the chosen discretisation as

$$\frac{m^k}{g} = \Delta x \Delta y \Delta p, \tag{15}$$

with $\Delta p$ in units of hPa and $\Delta x$, $\Delta y$ in units of km. Then, $\frac{m^k}{g}$ is set as parameter *partMass* (in hPa km$^2$ in the settings group *Lagranto* (Table A7), which is then internally in WaterSip converted to units of kg. For example, Sodemann et al. (2008b)
computed backward trajectories from a $50 \times 50$ km horizontal grid over Greenland, with a vertical spacing of 25 hPa for every 6 h time step over a given time period. Accordingly, each trajectory on this grid represents $680 \times 10^9$ kg of air arriving at a specific time interval.



### 2.4.2 Trajectory time interval $\Delta t$

Another important property of the input trajectory data for the WaterSip algorithm is temporal discretisation, i.e., the time
interval $\Delta t$ at which meteorological and position information from air parcel locations is available. Typically, $\Delta t$ corresponds
to the time interval of NWP boundary data used during the trajectory calculation. For global reanalyses a common interval is
6 h, but also 1 to 3 h time intervals have been used (Fremme et al., 2023). While trajectory location accuracy increases with a
higher input data frequency, there are also potential complications from higher time-resolution input data. As $\Delta q_c$ is well-tested
for $\Delta t = 6$ h with regard to separating signal from noise, smaller threshold values need to be used at shorter time intervals,
which do not necessarily have the same filtering effect (see Sec. 2.5). The trajectory time interval $\Delta t$ is set as parameter
*timeStep* (unit h) in the settings group *Case* (Table A1).

In a sensitivity study with WaterSip and FLEXPART using climate model data at $1.8 \times 1.8$ degree resolution as input, Fremme
et al. (2023) found that the time interval influenced the moisture source distance, with substantially smaller moisture source
distances for time intervals of less than 6 h. In that study, the authors then chose to compute trajectories at a time resolution of
1 h to obtain smaller position errors, while the WaterSip diagnostic was run at 6 h time interval to obtain the desired filtering of
interpolation noise. With such a combination of time intervals, a compromise can be obtained between trajectory accuracy and
sufficient separation between signal and noise in the WaterSip diagnostic.

### 2.4.3 Trajectory length $L$

One important consideration for computing trajectories is the trajectory length $L$, the duration over which the air parcel move-
ment is traced. Since trajectories become increasingly uncertain for longer calculation time, the trajectory length is an important
consideration for WaterSip applications (Stohl, 1998). In the approach of Stohl and James (2004), the results were strongly
dependent on the chosen integration time intervals. This choice introduces a subjective element in the analysis, since such
chosen time scales may not only vary with climate region, but also with seasons and weather systems. The accounting pro-
cedure implemented in WaterSip does not require a pre-defined trajectory length for its analysis. Past studies found that the
accounting algorithm provides regionally differing time scales, and typically finds only minor contributions beyond about 15
days (Winschall et al., 2014b; Laederach and Sodemann, 2016). The trajectory length is set as the number of time steps along
a trajectory as parameter *trajPoints* in the settings group *Case* (Table A1). One obtains the value of *trajPoints* from dividing $L$
by $\Delta t$. For example, 20-day trajectories with a 6-h time interval results in *trajPoints* set to 81.

From a global budget of atmospheric water vapour and precipitation fluxes, one obtains the often-cited global mean residence
time of water vapour of 8-10 days (e.g. Trenberth et al., 2007). Recent studies (based on Lagrangian moisture source diagnostics
with WaterSip) provide arguments that the residence time, or life time of water vapour, may be better described by a heavily
skewed life time distribution, that is characterised by a median value of about 4-5 days and a very long, thin tail at 20 days and
beyond (Sodemann, 2020; Gimeno et al., 2021). Except at high latitudes, where life times are particularly long, differences in
the source contributions are generally not sensitive to the trajectory length with the WaterSip algorithm beyond a trajectory
length of 15 days. Using the WaterSip diagnostic with trajectories of 10, 15, or 20 days length will therefore lead to relatively



less differing results that with the Stohl and James (2004) method. A generally recommended choice of trajectory length that also allows for the detection of long-range transport of water vapour is 20 days.

## 2.5 Control settings and threshold parameters

There are a number of parameter choices in the WaterSip algorithm that can be of critical importance for the results and their
interpretation. These choices are now discussed, starting with the choice of the target region, and then proceeding with the threshold settings that have a variety of impacts on the obtained results.

### 2.5.1 Target region and time period specification

A fundamental choice for an analysis with WaterSip is the geographic location of the target region. When used to diagnose precipitation origin, air parcels are retained for analysis if (i) they reside within the target region, (ii) during the preceding
time step had a specific humidity decrease larger than the precipitation threshold value (see Sec. 2.5.3), and (iii) had a relative humidity within chosen bounds (see Sec. 2.5.4). In the WaterSip configuration file, the target region is specified by a set of latitude and longitude bounds, namely the parameters *arrivalGridMinLon*, *arrivalGridMinLat*, *arrivalGridMaxLon*, *arrival-GridMaxLat* in the settings group *Grid* (Table A2). The grid spacing of the arrival grid in latitude, longitude increments, is defined by parameters *arrivalGridDx* and *arrivalGridDy* in the same settings group.

Moisture transport can be strongly modulated by topography, such as for example over Antarctica (Sodemann and Stohl, 2009). In order to focus the analysis on different altitude ranges within the target area, but also on selected elevation, several parameters are available in the settings group *Diagnostics*. Air parcels will only be retained for analysis if they arrive over topography with a height within the parameters *arrivalOroMin* and *arrivalOroMax*. Similarly, air parcels can be filtered according to their altitude above sea level using parameters *arrivalAltMin* and *arrivalAltMax*. In order to include all parcels
arriving within a domain, the parameters should be set to an height below zero (e.g., -100 m) for the lower, and a large height (e.g., 50000 m) for the upper threshold value. The used topography is ingested from the netCDF file specified at parameter *orographyFile* in settings group *Case*.

Another fundamental setting is the start and end time of the analysis, regulated by parameters *startDate* and *endDate* in section *Case*. Since the analysis proceeds backward from the time of arrival, the start date is larger than the end date. Importantly,
in order to obtain results for e.g. 10 day accounting along a trajectory, the time difference between start date and end date must be at least 10 days plus one time step. The length of the analysis period $L$ is defined by parameter *trajPoints* in settings group *Case*, and the duration of the entire accounting period for each trajectory is then given by the parameter *timeStep* times *trajPoints* (see Sec. 2.4.3).

### 2.5.2 Uptake threshold $\Delta q_c$

Specific humidity changes along air parcel trajectories are not only reflecting $e - p$, but also contain spurious noise. The reasons for such noise lie in the fact that positions along trajectories are interpolated from the gridded output of NWP models provided





at discrete times, as well as inaccuracies in the trajectory calculation itself (Stohl, 1998). There may also be an impact from parameterised atmospheric processes, such as convection and turbulence, that act at a time scale shorter than the trajectory time interval, and that are only detected by their residual effects (see Sec. 2.4.2).

In order to suppress or dampen such noise, the uptake threshold $\Delta q_c$ is introduced in the distinction of evaporation and precipitation events. Specifically, a moisture uptake is identified if $\Delta q > \Delta q_c$, while a moisture loss is correspondingly identified if $\Delta q < \Delta q_c$ (Fig. 1c). The uptake threshold is set as parameter *uptakeThreshold* in the settings group *Diagnostics* of the configuration file (Table A3). Thereby, the value must be given per trajectory time step (in $\mathrm{g\,kg^{-1}\,h^{-1}}$).

At a default value of $\Delta q_c = 0.2\,\mathrm{g\,kg^{-1}\,6\,h^{-1}}$ that is typically used in a broad range of mid-latitude conditions, only relatively
substantial uptake events are detected by the diagnostic. Tropical and sub-tropical regions may require a larger threshold value, whereas in colder and drier conditions, such as in polar or high-altitude regions, a lower threshold of $\Delta q_c = 0.1\,\mathrm{g\,kg^{-1}\,6\,h^{-1}}$ has been recommended (Sodemann and Stohl, 2009). A lower value for $\Delta q_c$ will lead to the inclusion of a larger part of the noise spectrum as moisture uptakes. A larger number of uptakes in turn leads to a higher chance of discounting previous moisture uptakes, and thus may induce a bias towards more local moisture sources.

As an alternative, WaterSip can be instructed to use a relative threshold value $\Delta q_{cr}$. In this case, noise is considered as a relative error, and thus the threshold value is computed at every time step as $\Delta q_c = q_i \cdot \Delta q_{cr}$. The relative threshold value is activated by setting the parameter *relativeThreshold*$= 1$ in the settings group *Diagnostics* (Table A3). The value of the uptake threshold is then interpreted as $\Delta q_{cr}$ (with unit 1). The impact of this relative threshold option has not yet been tested extensively for different climate zones and events.

In the context of the uptake threshold, it is interesting to consider the impact of data assimilation on the specific humidity of gridded NWP data, such as reanalysis data. Some data assimilation methods can introduce unphysical perturbations to the moisture field, the so-called assimilation increment. Assimilation increments are another reason to apply $\Delta q_c$ in the diagnostic. If trajectories are calculated using a model forecast without data assimilation, such as for climate model data, no assimilation increments are present in the gridded fields, and thus less noise would be expected in $\Delta q$. Fremme et al. (2023) investigated
the impact of different $\Delta q_c$ in CESM and NorESM climate model data compared to reanalysis data, and found quantitative but no clear qualitative differences in the result of the diagnostic that could be related to assimilation increments. Users should thus consider testing lower $\Delta q_c$ when using trajectories based on free-running model forecast data rather than reanalyses.

### 2.5.3   Precipitation threshold $\Delta q_p$

Similar to the uptake threshold, a corresponding threshold value exists for the identification of moisture losses due to pre-
cipitation along trajectories, the precipitation threshold $\Delta q_p$. As the same amount of numerical noise can lead to spurious identification of moisture losses as for moisture uptakes, the value of $\Delta q_p = -0.2\,\mathrm{g\,kg^{-1}\,6\,h^{-1}}$ is commonly used in mid-latitudes (Sodemann et al., 2008b), while a lower value such as $\Delta q_p = -0.1\,\mathrm{g\,kg^{-1}\,6\,h^{-1}}$ appears more suitable for polar regions (Sodemann and Stohl, 2009). In general, it is recommended for reasons of consistency to use symmetric settings for the uptake and precipitation thresholds. The precipitation threshold is set as parameter *precipThreshold* in the the settings group
*Diagnostics* of the configuration file (Table A3). As the detection of moisture losses along trajectories leads to a discount of the



remaining contribution from earlier uptake events, a lower precipitation thresholds can lead to relatively less remote moisture sources. The parameter *relativeThreshold* also affects the interpretation of the value in *precipThreshold* as absolute or relative parameter value.

### 2.5.4 Critical relative humidity threshold $\mathrm{RH}_c$

The Lagrangian precipitation estimate $\tilde{P}$ is based on moisture losses during the last time step of arriving trajectories. In order to further support that a detected moisture loss is due to cloud processes, the critical relative humidity threshold $\mathrm{RH}_c$ is introduced. If $\mathrm{RH}_0 > \mathrm{RH}_c$ or $\mathrm{RH}_1 > \mathrm{RH}_c$, the air parcel is associated with clouds during arrival, and thus any negative $\Delta q$ is assumed to be due to precipitation. The critical relative humidity threshold is set as parameter *arrivalRHMin* in the the settings group *Diagnostics* of the configuration file (Table A3). The default value for $\mathrm{RH}_c$ is 80 (in units of %), corresponding to a

common threshold value in parameterisations of sub-grid cloud cover. Several studies indicate that depending on latitude and predominant weather regime, the Lagrangian precipitation estimate $\tilde{P}$ increases with lowering of $RH_c$. Such lower threshold values may be argued for in particular in mixed-phase and ice cloud regimes (Terpstra et al., 2021), and in the case of predominantly convective precipitation, that are not necessarily associated with grid-scale cloud cover (Fremme and Sodemann, 2019). An additional parameter *arrivalRHMax* in the settings group *Diagnostics* allows to focus the analysis on a a specified range of

RH during air parcel arrival (Table A3).

Note that a decrease in specific humidity, detected as a moisture loss, can also be a consequence of mixing with drier air, for example due to dry convection. Moisture loss events that take place in an unsaturated environment, i.e., when $\mathrm{RH} < \mathrm{RH}_c$ at both time steps used to calculate any $\Delta q_i$ are therefore counted and processed as dry mixing events, rather than precipitation events (see Sec. 2.5.3). While these mixing events are written to a different output variable (see Sec. 4.1), the consequences for

the accounting of moisture losses is the same for saturated and unsaturated conditions.

### 2.5.5 Boundary-layer height scale factor $s_h$

Another important parameter is the distinction between uptake events that are detected when the air parcel is within the boundary layer, or above. Since air parcel trajectories are 3-dimensional, the uptakes are identified when the partcels are at a given height above ground. Evaporation from the surface is assumed to be most plausibly contributing to the moisture increase

in an air parcel location if it is within or close to the turbulently mixed atmospheric boundary layer. This resembles the "footprint analysis" of the lowermost 100–300 m above ground, which is commonly used in atmospheric chemistry applications (e.g., Hirdman et al., 2010). Detrainment and venting of boundary-layer air to the nearby free troposphere is another likely pathway for producing moisture uptakes close to the boundary-layer top. Therefore, and to take into account diurnal boundary-layer height variability and the general uncertainty of boundary-layer height calculations, a boundary-layer height scale factor $s_h$

with a default value of $s_h = 1.5$ is introduced in the diagnostic. All moisture uptakes that are identified when height $z \leq s_h \cdot z_h$ (with $z_h$ the current boundary layer height) are assigned to the output field category of boundary-layer moisture uptake, while all other uptakes are in the free-tropospheric moisture uptake category (see Sec. 4.1). The distinction between these two





categories is also carried over to the accounted fractions (Sec. 4.3), and the forward projection of different quantities (Sec. 4.4). The relative threshold value is set as parameter *blhScale*= 1.5 in the settings group *Diagnostics* (Table A3).

Studies with the WaterSip methodology have used $s_h$ in different ways. While Sodemann et al. (2008b) introduced the parameter to obtain more reliable results, the free-troposphere and boundary-layer uptakes have since been observed to commonly overlap spatially (e.g., Sodemann and Zubler, 2010). In a detailed investigation of the atmospheric conditions associated with free-troposphere uptakes in the Mediterranean, Winschall et al. (2014a) found that these were connected to the deep venting of boundary-layer air to the free troposphere due to moist convection. Furthermore, free-troposphere uptakes are generally the

lesser fraction of the total uptakes, typically on the order of 20–40% (Sodemann and Zubler, 2010; Laederach and Sodemann, 2016). Several studies have therefore in the past considered both uptake categories jointly, in particular in warmer climates (e.g., Martius et al., 2012; Fremme and Sodemann, 2019; Fremme et al., 2023). Users will have to ultimately decide according to their scientific objectives how to apply $s_h$, and how to make use of the boundary-layer and free-troposphere uptake categories.

### 345  2.5.6   Output grids and gridding radius $r_g$

WaterSip uses the mass represented by each air parcel to transfer the results of the diagnostic onto a grid which will then be written to an output file for further use. There are two output grids in WaterSip. The coarser source and transport grid covers the entire potential source regions of the traced moisture, and is typically of hemispheric or global dimensions. The typically smaller arrival grid covers the target region of the analysis, and has typically a finer grid spacing to reveal spatial details within

the arrival domain.

    Both grids are specified by a respective set of parameters in the setting group *Grids* (Table A2). The source grid is defined by the parameters *sourceGridMinLon, sourceGridMaxLon, sourceGridMinLat, sourceGridMaxLat*, which provides the boundaries in latitude-longitude coordinates. The parameters *sourceGridDx, sourceGridDy* set the grid spacing in longitude and latitude direction, respectively. A corresponding set of grid parameters exists for the arrival grid (*arrivalGridMinLon,*

*arrivalGridMaxLon, arrivalGridMinLat, arrivalGridMaxLat, arrivalGridDx, arrivalGridDy*).

    Quantities are gridded to both grids using weighting factors. The weight for gridding a quantity $\xi$, for example the air temperature at the time of arrival $\Delta q_0^k$ is obtained from the water mass lost from an arriving air parcel $k$ as

$$m_w^k = m^k \cdot \Delta q_0^k. \tag{16}$$

With $K$ trajectories arriving as a vertical stack (Fig. 1b), the weighted average of all $\xi^k$ at a location $\lambda, \phi$ is thus obtained as

$$\tilde{\xi}(\lambda, \phi) = \frac{\sum_{k=1}^{K} \xi^k \cdot m_w^k}{\sum m_w^k}. \tag{17}$$

    In order to transfer this arrival point information onto the arrival grid, a representativeness of this information is assumed. WaterSip uses a circular gridding radius $r_g$ to transfer air parcel quantities $\xi$ onto the arrival grid. For example, with trajectories





computed at a horizontal distance of $50 \times 50$ km, quantities can be assumed to be representative for a circular area with radius 35 km. If the arrival grid for this setup is then chosen to be *arrivalGridDx=arrivalGridDy*$= 0.25°$, and the gridding radius

is set to $r_g = 40$ km, one can expect to retain any finer interpolated structures within the arrival domain. The gridding radius is set as parameters *arrivalGridRadius* in settings group *Grids*. Corresponding reasoning applies to the source grid radius *sourceGridRadius*. Here, a common combination for a hemispheric output grid is *sourceGridDx=sourceGridDy*$= 1.0°$, and *sourceGridRadius*$= 110$ km.

During the gridding, it is tested how a circular gridding area overlaps with each grid box on the grid, and a corresponding

fraction of the total quantity to be gridded is then added to each grid box (Škerlak et al., 2014). It is important to be aware what impact user choices have on the results. A too small gridding radius for a given grid will lead to information gaps between grid points, while a too large gridding radius will cause smoothing and potentially loss of detail in the gridded fields.

A final choice regarding the gridding is whether uptake events are gridded with the time stamp of the arrival in the target region, or to the time when uptakes take place. The choice of the assigned time is important for later analysis of the results.

In the first case, moisture sources are valid for all precipitation arriving within a chosen time period in the target area, and can be compared to other quantities regarding the arrival time. In the second case, the moisture sources can be compared to properties at the source regions during valid time, for example total evaporation fields. The assignment time is set as parameter *assignToUptake* in settings group *Grids*, where 0 denotes that the uptakes are assigned to arrival time, and 1 denotes assignment to uptake time.

### 2.5.7    Water vapour diagnostics

In addition to tracing back water vapour that leads to precipitation, WaterSip also allows to identify the source and transport properties of water vapour that resides in the atmosphere within a target area at a given time. If WaterSip is used as a water vapour source and transport diagnostic, rather than a precipitation diagnostic, the gridding weight of each traced air parcel is obtained from the total water mass $m_w$ of an air parcel of trajectory $k$ at arrival time $t = 0$, computed as

$$m_w^k = q_0^k \cdot m^k. \tag{18}$$

This water mass $m_w^k$ is then used to weight different quantities during gridding, instead of the loss of water mass during the last time step, i.e., $\Delta q_0^k \cdot m^k$ (Eq. 16). Apart from this change, the moisture source diagnostic works in the same way when tracing the sources and properties of precipitation water. The water vapour diagnostic option allows for instance to identify the transport properties of water vapour in regions where no precipitation occurs. The water vapour diagnostic is activated by

setting parameter *analyzeVapour* as 1 in the setting group *Diagnostic* (Table A3).

### 2.5.8    Other available moisture source diagnostics

To enable comparison of the Sodemann et al. (2008b) algorithm to other approaches, several additional moisture source diagnostics have been implemented into the software tool. These include the e-p diagnostic by Stohl and James (2004), in which all





particles arriving in a region and precipitating are gridded without further distinction according to vertical location, and with-
out accounting. Another diagnostic is the method of Gustafsson et al. (2010), in which the last location where the trajectory
gained/lost in specific humidity is gridded. Furthermore, the method of Dirmeyer and Brubaker (1999) is available (modified
by an accounting step). In order for this last method to work, surface evaporation needs to be available on the trajectory input
data. The method of Stohl and James (2004) is included by default in the output files. The other two diagnostic methods are
available by editing the call to the corresponding processing routines in the class *Trajectory* in file `Trajectory.cpp` the
C++ code.

## 3 Input data, run configuration, and output data

In order to run the WaterSip diagnostic, a set of suitable input data needs to be available. In this preparatory step of the moisture
source analysis, users need to make are a number of necessary and optional choices as detailed below. Running the WaterSip
tool then produces a set of output files in netCDF format, which are also introduced in this section.

### 3.1 Creating and reading trajectory input data

The setup for creating trajectory input data to WaterSip varies technically depending on whether a particle dispersion model
(FLEXPART) or an air mass trajectory model (LAGRANTO) is used. There are also some general considerations to be made
such that the WaterSip diagnostic output becomes representative for the chosen target area.

Commonly, hemispheric or global domain-filling forward simulation are set up with the particle dispersion model FLEX-
PART. In domain-filling mode, FLEXPART automatically discretises the air mass over the chosen target area into a specified
number of particles of equal mass, and distributed according to pressure. A larger number of particles will provide statistically
more robust results, in particular for more long-range and long-time water vapour transport. Particle trajectories are run forward
indefinitely for the simulation time. WaterSip then extracts a sub-set of these representative trajectories as given by the run pa-
rameters. Such a simulation setup has also been applied in the example case presented here. So-called particle dump needs to
be activated in FLEXPART to obtain the input data needed by WaterSip. Writing out additional variables with each particle
position requires modification of the routine `partoutput_pos.f90` or `partoutput_short.f90` of the FLEXPART
code. An example for this modification is available in the online supplement (Sodemann, 2025). Running FLEXPART in back-
ward mode can be a more efficient alternative to a global domain-filling run, but requires a new release interval to be defined
for every time step. General instructions for setting up and running FLEXPART are available in the respective literature (Stohl
et al., 2005; Pisso et al., 2019; Bakels et al., 2024).

When used together with FLEXPART input, the WaterSip configuration file needs to contain a section *Flexpart*, where
several parameters need to be specified (Table A6. The parameter *particleMass* contains the mass per particle as either specified
in the FLEXPART releases file, or as is written out by FLEXPART in domain filling mode in the output file *part_mass*. The
input files need to be contained in a folder specified by parameter *inputDir* in section Case in the configuration file. As





individual partoutput files are read in from FLEXPART output for each time step, the pathname is specified with a wild card pattern representing the date string, e.g. *inputDir = /path/to/directory/partoutput_pos_%s*.

When using output from a trajectory model, such as LAGRANTO (Sprenger and Wernli, 2015), the air mass over the target area needs to be discretized manually. This is achieved by defining the start locations of the trajectory calculations in LAGRANTO's `startf` file. One common way to discretise a target region is to define a regular distance spacing in projected coordinates (e.g., one starting point every 25 or 50 km in both directions). In the vertical, it is recommended to space the starting points at equal pressure intervals. This way, each air parcel trajectory represents an equal amount of mass, which can be entered in the configuration file for a LAGRANTO setup according to Eq. 15.

It is recommended to run LAGRANTO program `caltra` with the flag '-j' to make trajectories avoid topography. WaterSip currently only works with the text-based output format of LAGRANTO. Since trajectory variables such as air temperature and specific humidity can be at any column in these files, the column number for each variable needs to be specified explicitly in the configuration file as parameters (Table A7. The input path for LAGRANTO trajectory files is again specified using parameter *inputDir* in section *Case* of the configuration file, using a file name pattern with placeholder `%s` to match trajectory files to the desired date range.

## 3.2 Running the WaterSip diagnostic

The WaterSip code is written in the programming language C++. A sample makefile is included with the code repository to compile the program code, and that users may adapt to their computational environment. An installed netCDF and netCDF-C++ library are in addition required to compile the program code. Lightweight parallelisation has been included by means of OpenMP directives in the code. Compiling with OpenMP is optional.

After compiling WaterSip, the input files need to be made available from an external source, such as running FLEXPART or LAGRANTO. The directory names for input and output data need to be created and specified in the configuration file. Then, all other configuration file settings, such as the time period, the target region, diagnostic parameters, need to be set. Importantly, the sections in the configuration file specific for the used input data need to be included and edited to match the input data (Table A6 and A7).

To start a diagnostic run, the compiled WaterSip program is then executed from the command line. The name of the configuration file needs to be provided as only parameter when WaterSip is started. WaterSip then starts executing, and provides either output to the terminal about the progress of the analysis, including a time estimate until completion, or an error message that will give an indication how the setup and configuration file need to be adjusted.

The input data and configuration file for the test case in Scandinavia presented throughout this manuscript is available as a supplement to this manuscript (Sodemann, 2025). Users are invited to experiment with the test case before creating their own setup to get familiar with the usage and output from WaterSip.





## 3.3 Output files

When a diagnostic run completes, a set of output files is written to the file location specified as parameter *outputDir* in the configuration file section *Case*. The main output files, containing gridded fields and time series for different averaging times (time step, daily, monthly, annually, all), are created in netCDF format (Table 2). In particular the grid files for days and time steps can become rather large for longer runs. The creation of gridded output files for selected averaging times can therefore be 460 deactivated by setting the respective parameters in parameter set *Output* from 1 (true) to 0 (false, see Table 2). The series files contain time averages at different intervals for the different gridded fields, characterized by the precipitation-weighted mean, the unweighted standard deviation, and the minimum and maximum values. The sectorisation results are also contained in the series files (Sec. 4.7). Further output files comprise histogram files (Sec. 4.8) and trajectory files (Sec. 4.9).

465 Which variables are contained in the output files can be controlled by the parameters in section *Variables* in the configuration file (Table A5). Output variables are included in the output by setting the corresponding parameter values from 0 to 1. Users can also chose to include static fields, such as topography or the land-sea mask using parameter *staticFields*. The choice in the section *Variables* affects both grid and series files. If output is activated for variables that are not contained in the input file, WaterSip calculates these from the available information if possible, and otherwise fills in missing values during output.

**Table 2.** Summary of the available output files in WaterSip, including file format, and corresponding parameters in the configuration file.

| Name | Content | Format | Filename |
|---|---|---|---|
| Grid files | Spatial properties of moisture sources and arrival regions | netCDF | \<CASE\>_grd_(all\|year\|month\|day\|step).nc |
| Series files | Mean/min/max/std of regionally and globally averaged moisture source and arrival region properties | netCDF | \<CASE\>_ser_(all\|year\|month\|day\|step).nc |
| Histogram files | Histograms of select quantities identified for each time step along each trajectory | text | \<CASE\>_hst_steps |
| Trajectory files | Air parcel trajectories for identified precipitation events | binary (.traj) | \<CASE\>_\<DATE\>(.traj) |

## 3.4 Example run configuration

470

In order to exemplify the use of the WaterSip diagnostic, an example run configuration has been set up for a domain over Scandinavia in Northern Europe. To this end, the FLEXPART particle dispersion model in Version 10.4 has been run with input fields from the ERA5 reanalysis for a northern hemisphere domain with 10'000'000 particles in domain filling mode. The simulation was initialised on 20 July 2022 and run forward for 31 days until 20 August 2022. FLEXPART particle output 475 files contained latitude, longitude, height, air temperature, specific humidity, boundary-layer height, and air density at each particle's location and 2m air temperature, 2m specific humidity and topography height at the surface location below the particle. WaterSip was then configured to identify the moisture source and transport properties for a time period of 10 days



(10 to 20 August 2022) in the region 54–71°N and 4–32°E. A 6h time step and threshold parameter values $RH_c = 80$ and $\Delta q_c = 0.2$ gkg$^{-1}$6hr$^{-1}$ were used for identifying moisture uptakes and losses along the trajectories. Output was activated for all available variables and time steps. The WaterSip setup and output files setup for this example run are available on the code repository (Sodemann, 2025).

## 4    Moisture source, transport and arrival conditions

Now that the factors influencing how the WaterSip diagnostic operates have been presented, what follows is a description of the different output fields that quantify conditions at the moisture source, during moisture transport, and during the arrival of the moisture as part of the output. The results are thereby exemplified using the case study setup over Scandinavia as described above.

### 4.1    Source grid quantities

The most fundamental result of the moisture source diagnostic are the moisture sources, that is the spatial distribution of the moisture uptakes that contribute to precipitation in the target region. An example for the total moisture sources is shown for the arrival domain covering Scandinavia (Fig. 2a, yellow frame). This moisture source area may be interpreted as the subset of all evaporation at the moisture sources that ends up as precipitation in the target region. The units of kg m$^{-2}$ d$^{-1}$, also corresponding to mm d$^{-1}$, are obtained from gridding the weighted specific humidity uptakes for trajectories precipitating in the target region. When integrated spatially, taking the area of each source grid box into account, one obtains the total precipitation mass in the target area during this time interval. If divided by the target area, the precipitation rate for the analysis period then corresponds exactly to the value of the Lagrangian precipitation estimate.

In the example case, the total moisture source has its larges values over southern Scandinavia, also stretching into Central and Eastern Europe (Fig. 2a, purple, green and red areas). In addition, a large area is covered by moisture uptakes with 0.25 mm dy$^{-1}$ (blue shading), even reaching the East Coast of North America. In order to obtain a valid interpretation, it is important to consider the amount of moisture uptake in conjunction with the respective covered area. In the example here, the gridding radius on the global maps with a 1.0×1.0° spacing was set to 120 km (see Sec. 2.5.6). A coarser grid spacing and a larger gridding radius would give more smoothed output fields.

The interpretation of moisture source maps can be greatly facilitated in a percentile representation (Fremme and Sodemann, 2019; Fremme et al., 2023). This can be obtained by determining, for example, the 50th and 80th percentiles of the accumulated moisture source contributions, sorted from largest to smallest grid point values. Such a percentile representation is particularly helpful to compare the shape of a moisture source footprint across different events or seasons. In the example for the present case, the 50th and 80th percentile contours for the total moisture source footprint (Fig. 2a, dashed and dotted red lines) highlight that almost half of the precipitation is contributed by areas bounded by the orange shading (contour value 0.09 mm dy$^{-1}$), and the large majority is bound by light blue area (contour value 0.24 mm dy$^{-1}$).



The moisture source detection algorithm distinguishes between uptakes that take place within or close to the boundary layer top, and such that are in the free troposphere, above the boundary layer. For the example case, the moisture sources in the boundary layer dominate (Fig. 2b), while the free-troposphere moisture sources supplement the boundary-layer uptakes over the North Atlantic and Central Europe (Fig. 2c). In particular the uptakes above Central Europe for this summertime case may be a reflection of convective detrainment of moist boundary layer air to the free troposphere (Sodemann and Zubler, 2010; Winschall et al., 2014a).

The moisture source footprint is here compared to the result of the Stohl and James (2004) method for the same case (Fig. 2d). The vertical integral of $e - p$ of all the particle trajectories over 10 days (the quantity $(e - p)_{10\,d}$ introduced by Stohl and James (2004)) shows patches where evaporation dominates (red shading) and where precipitation dominates (blue shading). While evaporation regions dominate in further away, precipitation overcompensates closer to and over the arrival domain, masking potential evaporation from these regions. The presence of such compensating effects, as well as the strong dependency on a selected time scale (not illustrated here), are two of the most important differences compared to the results from the WaterSip algorithm. The red contours, indicating the 50th and 80th percentile of total moisture sources from WaterSip, further underline the differences between the results from the two methods.

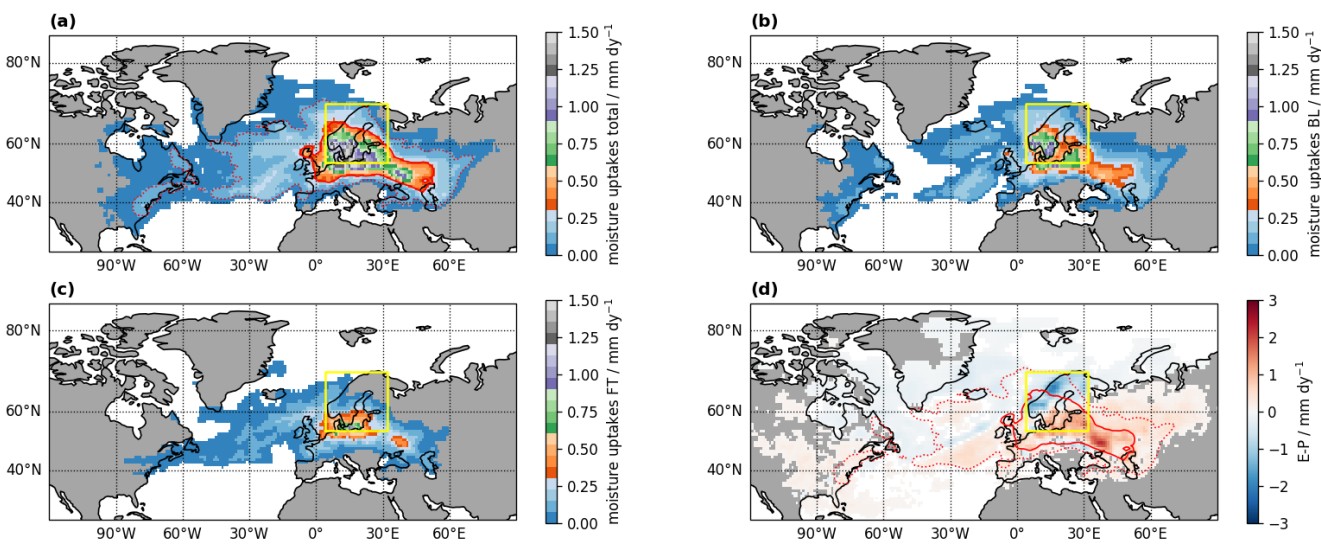

**Figure 2.** Source grid quantities identified for the case study period in 2022. (a) Total moisture uptakes (shading, mm d$^{-1}$), (b) Boundary-layer moisture uptakes (shading, mm d$^{-1}$), (c) Free-troposphere moisture uptakes (shading, mm d$^{-1}$), (d) Lagrangian evaporation minus precipitation budget over 10 days, following Stohl et al. (2005) (($e - p)_{10\,d}$ in mm d$^{-1}$, shading). Red contours indicate the 50th (solid) and 80th (dotted) percentile of the total moisture uptakes from WaterSip. Yellow box denotes the target region over Scandinavia.

In addition to providing the moisture source footprint for an entire analysis period, the WaterSip diagnostic provides gridded moisture sources for each time step in the steps grid file (Sec. 3.3 and Table 2). At 06 UTC on 10 Aug 2022, the precipitation





in the target area was sourced from a rather large banded region stretching across the North Atlantic, and reaching into Eastern North America (Fig. 3a). This map shows the moisture uptakes regarding the time of arrival in the target domain. If one wants to compare the moisture uptakes directly with, for example, gridded evaporation fields, one can use the parameter *assignToUptake* (Sec. 2.5.6). The corresponding plot for uptakes on 06 UTC at 10 Aug 2022 shows all evaporating moisture at the synoptically valid time. In the given example, relevant uptakes at that time are detected south of Iceland, over Scandinavia, and over western Russia (Fig. 3b). The moisture from these uptakes will lead to precipitation over the target area at a later time.

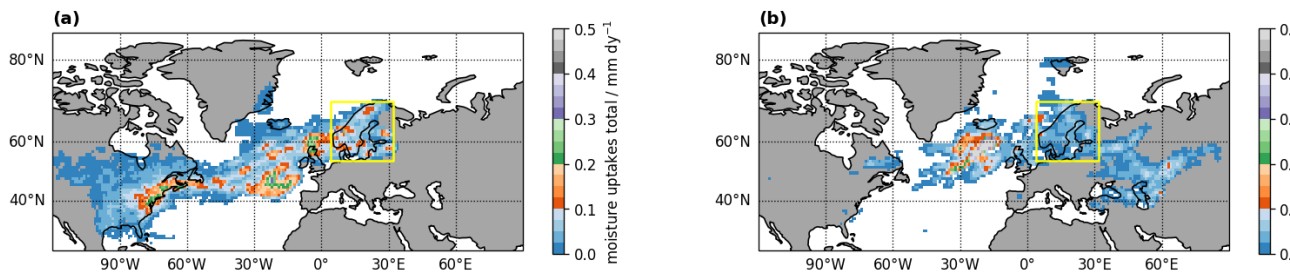

**Figure 3.** Instantaneous snapshot of source grid quantities identified during the case study period with different assignment time. (a) Total moisture uptakes arriving in the target area on 06 UTC at 10 Aug 2022 (shading, mm d$^{-1}$), (b) Total moisture uptakes taking place on 06 UTC at 10 Aug 2022 (shading, mm d$^{-1}$). Yellow box denotes the target region over Scandinavia.

## 4.2  Transport grid quantities

A number of additional quantities that are indicative of the moisture transport conditions can be obtained on the same grid as the moisture source information. These transport-related grid quantities can support the interpretation of the obtained moisture sources in various ways.

The transport pathways of air parcels towards the target region are visible from the air parcel density (Fig. 4a). The air parcel density is an integral of all particles bound to the target region at a given time interval. The example case shows the relatively large overall area from where air masses originate, and the important role of Greenland topography for limiting the air parcel transport. The quantity moisture transport provides information about the water vapour movement in these air masses towards the target area. The moisture transport is obtained as the gridded product of the specific humidity in tracked air parcels on the way to the target area, irrespective of their sources. For the example case, the moisture transport map highlights a dense area of moist air south of the target area, and where the moist air crosses the North Atlantic and advances towards the target area (Fig. 4b, mm day$^{-1}$). In combination, these two transport quantities allow one to assess the main pathways of air mass transport and the differences to moist air advection to the target region.

Decreases of specific humidity during transport can be due to either precipitation events or mixing. The air mass mixing and moisture rainout contain this gridded information as a result of the diagnostic (Sec. 2). For the present example case, air mass



mixing is mostly identified over continental regions of eastern North America and over Central and Eastern Europe (Fig. 2c). In comparison, moisture rainout is concentrated over the arrival domain, and narrow transport pathways over the North Atlantic and across Central Europe (Fig. 2d), related to precipitation formation in air masses that ascend as they approach the target area.

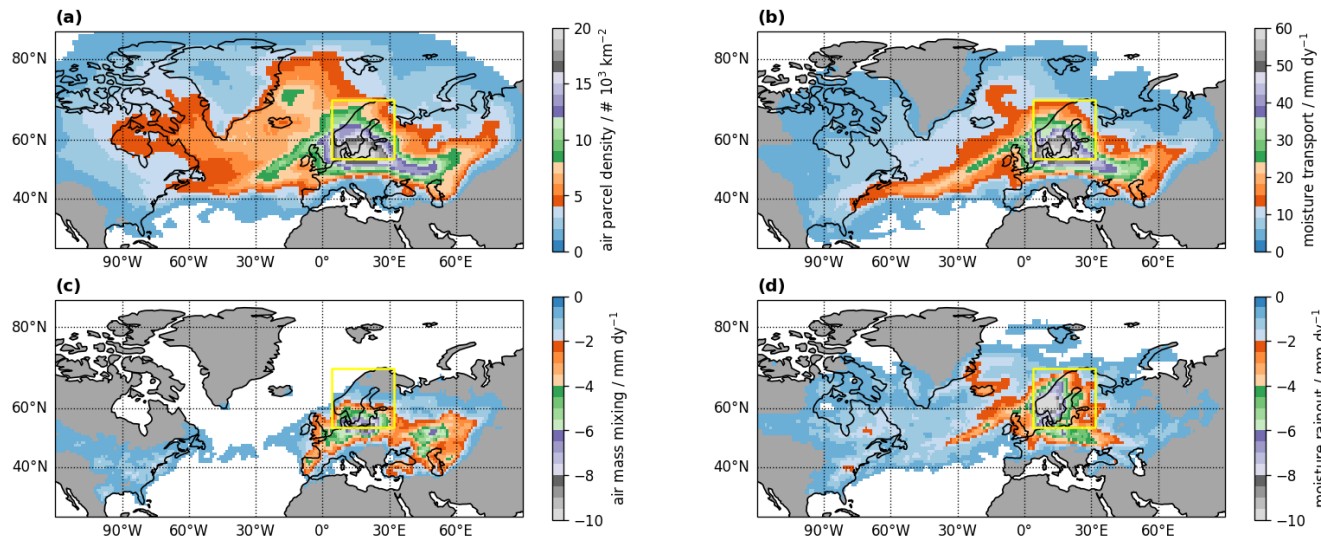

**Figure 4.** Transport grid quantities identified within air masses bound to the target region for the case study period. (a) Air parcel density (shading, $m^{-2}$), (b) moisture transport (shading, mm $d^{-1}$), (c) air mass mixing (shading, mm $d^{-1}$), (d) moisture rainout (shading, mm $d^{-1}$). Yellow box denotes the target region over Scandinavia.

## 4.3 Moisture arrival quantities

Depending on the size of the arrival domain (or target area), it can be interesting to spatially resolve differences moisture sources and transport within the chosen region. WaterSip quantities that are connected to the arrival time of air parcel trajectories, and which are gridded onto the arrival grid, are termed here *arrival quantities*.

One of the most important arrival quantity is the Lagrangian precipitation estimate $\tilde{P}$ (Sec. 2.2). In the example case, there are clear distinctions between the coastal regions of western Scandinavia with an estimated precipitation of above 6 mm $d^{-1}$, and large parts of the Baltic Sea, southern Sweden and Finland with below 2 mm $d^{-1}$ during the case study period of 10-20 Aug 2022 (Fig. 5a). Regions with high $\tilde{P}$ approximately correspond to locations where more air parcels arrive with precipitation, as reflected by the arrival count (Fig. 5d). In regions where only few air parcels contribute to gridded quantities, gridded results may show large spatial variations that are not statistically robust. The arrival count can be used for masking such grid points.




Another set of important arrival quantities are the accounted fractions (Sec. 2.3). The total accounted fraction in the example case is uniformly above 95% (Fig. 5e). Subdividing this quantity into the fractions accounted for by boundary layer uptakes (Fig. 5b) and free troposphere uptakes (Fig. 5c) shows a uniform split into approximately 60% boundary layer and 40% free troposphere contributions. These shares appear consistent with the corresponding moisture source maps (Fig. 2b, c).

In addition, the thermodynamic conditions of water vapour at the time of arrival over the target domain can be gridded onto the arrival grid. For example, the arrival temperature is the temperature of each air parcel at arrival, gridded using weights corresponding to the precipitation estimate at arrival grid location $(\lambda, \phi)$ for the total accounted fraction, $\tilde{P}_{\text{tot}}$:

$$T_{\text{arr}}(\lambda, \phi) = \frac{\sum_{k=1}^{K} T_0^k \cdot f_{\text{tot}}^k \cdot \Delta q_0^k \cdot m^k}{\tilde{P}_{\text{tot}}^k} \tag{19}$$

In the present example, arrival temperatures vary between about 2 and 8 °C (Fig. 5f). Thereby, regions where least precipitation fell have arrival temperatures that are lowest, with -2°C. Correspondingly, the air pressure at arrival is also available as gridded quantity (Table C2, not shown).

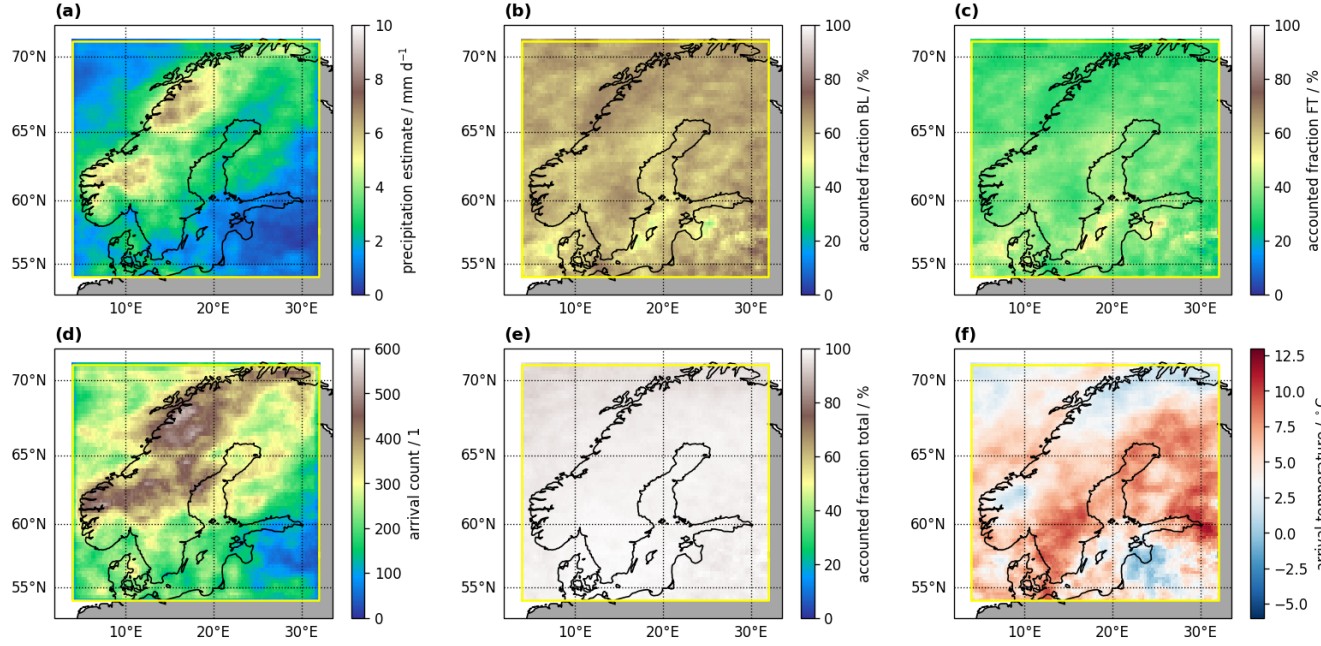

**Figure 5.** Arrival quantities identified for the example case in 2022. (a) Lagrangian precipitation estimate $\tilde{P}$ (mm d$^{-1}$), (b) accounted fraction of $\tilde{P}$ from boundary-layer uptakes (shading, %), (c) accounted fraction of $\tilde{P}$ from free-troposphere uptakes (shading, %), (d) arrival count (shading, 1), (e) total accounted fraction of $\tilde{P}$ (shading, %), (f) arrival temperature (shading, °C). Quantities in (b)-(f) are weighted averages, using the specific humidity decrease during the last time step before arrival as a weight.



## 4.4 Lagrangian forward projection of moisture source quantities

As WaterSip identifies the location of a moisture source, different properties of the source region can be computed, such as the sea surface temperature, or its latitude and longitude. In particular for larger arrival regions, and such that contain geographic or climatic gradients, the source region location and its properties may vary substantially within the arrival domain. In order to make such spatial differences in the source region properties visible, WaterSip provides Lagrangian forward projections (LFPs) of the source region properties on a spatial grid covering the arrival domain.

LFPs are obtained as the weighted average of a given moisture source property for each trajectory. The weighted average is then gridded on the arrival grid, weighted by $\tilde{P}$ (or by the air parcel specific humidity in the case of water vapour diagnostics). Thus, for each trajectory $k$, a moisture source or transport quantity $\zeta$ is transferred from the geographic uptake location $(\phi_u, \theta_u)$ to the geographic location at the arrival region $(\phi_a, \theta_a)$ using the fractional contributions $f_i^k$ obtained from the accounting step as a weight, and resulting in LFP quantity $\tilde{\zeta}$ for air parcel $k$:

$$\tilde{\zeta}^k(\phi_a, \theta_a) = \frac{\sum_{i=1}^{M} \zeta^k(\phi_u, \theta_u) \cdot f_i^k \cdot \Delta q_i^k \cdot m^k}{\sum_{i=1}^{M} f_i^k \cdot m^k} \tag{20}$$

The different $\tilde{zeta}^k$ are then again gridded using Eq. 19. In creating LFPs, an important choice is whether the total accounted fraction of the precipitation estimate is used in the gridding, or only the boundary layer or free troposphere fraction. Parameter *forwardProjectionMode* in parameter section *Diagnostics* determines this choice (1: boundary layer, 2: free troposphere, 3: total fraction).

For the case of Scandinavia, the mass-weighted moisture source latitude (centroid latitude) ranges from 42–58 °N, with a clear latitudinal gradient (Fig. 6a). Corresponding moisture source longitudes range from -60 to 40°E (Fig. 6b). In combination, the Baltic states stand out with more easterly and southerly moisture source that other regions of the arrival domain. Such a forward projection of source region properties thus provides insight into the spatial variability within the arrival domain. When used on large (global) domains, Lagrangian forward projections can provide interesting hydro-climatic insights (Laederach and Sodemann, 2016; Sodemann, 2020).

The source land fraction (Fig. 6d) emphasises that the Baltic states are in this case dominated by land uptakes (source land fraction >80%), while sources are increasingly oceanic towards the northwest of the domain (<30%). The fraction of land sources is obtained from evaluating a land-sea mask at the locations below the moisture uptakes. In case the mask files contains fractional land cover information, values above a threshold of $> 0.5$ will classify the source location as land.

The source distance map for the example case shows a similar pattern as the source longitude, with more nearby sources (<1500 km) over Denmark and southern Sweden in the southwest, and patches of more long-distant transport (>3500 km) in the north of the domain (Fig. 6c). The distance of the moisture sources is computed as a spherical distance for each uptake location and then gridded according to Eq. 20.

The moisture source temperature is obtained here from the 2m air temperature ($T_{2m}$) over land and ocean regions (Fig. 6e). For the present case, moisture source temperatures are relatively uniform, with warmest source temperatures in regions where





land sources dominate. Additionally, the LFP of moisture source relative humidity, air pressure at the sources, surface skin temperature, and specific humidity at 2m ($q_{2m}$) at the sources may be available, depending on the what has been written out

to the FLEXPART and LAGRANTO input files. Further LFP quantities characterises the thermodynamic conditions at the moisture source (Table C1). In the case that source skin temperature and $q_{2m}$ or RH and $T_{2m}$ are available an approximation for the stable water isotope parameter d-excess at the moisture sources can be computed from the relation of Pfahl and Sodemann (2014):

$$d_{\text{src}} = -0.54\%o\,\%^{-1} \cdot \text{RH}_{\text{SST}} + 48.2\%o \tag{21}$$

Thereby, the relative humidity with respect to SST ($\text{RH}_{\text{SST}}$) is obtained using saturation vapour pressure according to Flatau et al. (1992). For the example case, the computed d-excess shows a gradient that mirrors land properties, with higher values (15–20 ‰) where land sources dominate, while ocean source dominated regions show a lower d-excess, slightly below the global average in precipitation of 10 ‰ (Araguás-Araguás et al., 2000). The d-excess parameter is both an interesting diagnostic for paleoclimatic studies, and can be used to relate moisture transport to water isotope measurements on meteorological time scales

(e.g., Weng et al., 2021).

### 4.5   Lagrangian forward projection of moisture transport quantities

In the a similar way as moisture source quantities (Eq. 20), it is possible to project transport quantities onto the arrival region, with the exception that the average is obtained from the current accounted fraction during the moisture transport. For example, $\tilde{\xi}^k$, the forward projected moisture transport temperature valid during arrival of air parcel $k$ is obtained by averaging the air

parcel quantity $\xi^k$ along the entire transport path from 1 to $N$, using $f_i^k$, the explained fraction at step $i$, as a weight:

$$\tilde{\xi}^k(\phi_a, \theta_a) = \frac{\sum_{i=1}^{N} \xi_i^k \cdot f_i^k \cdot m^k}{\sum_{i=1}^{N} f_i^k \cdot m^k} \tag{22}$$

The LFP of the moisture transport temperature obtained this way shows relatively warm transport for the majority of air masses at around 10°C (Fig. 7a). In comparison, the condensation temperature, computed as transport temperature conditioned on RH> 80°C, shows that saturation is reached at colder and more variable temperatures throughout the domain, including

below-zero temperatures in the southeast (Fig. 7b). Comparison to the transport pressure shows corresponding lower-level transport in the regions of warmer condensation temperature, with air pressure of 850 hPa and above in Northern Scandinavia, compared to ~700 hPa over the Baltic states (Fig. 7c).

A quantity that allows to inspect the time scale of moisture transport is the transport time, obtained as the difference between the time of each uptake event and the corresponding arrival time. For the example case, the transport time shows the larges

values with above 5 days in northern Scandinavia, whereas Denmark and southern Sweden have transport times of 3 days and below (Fig. 7d). The transport time in this case thus bears similarities with the transport distance (Fig. 7e) and the source distance map (Fig. 6c). In contrast to the moisture source distance, the transport distance is obtained as the total distance along



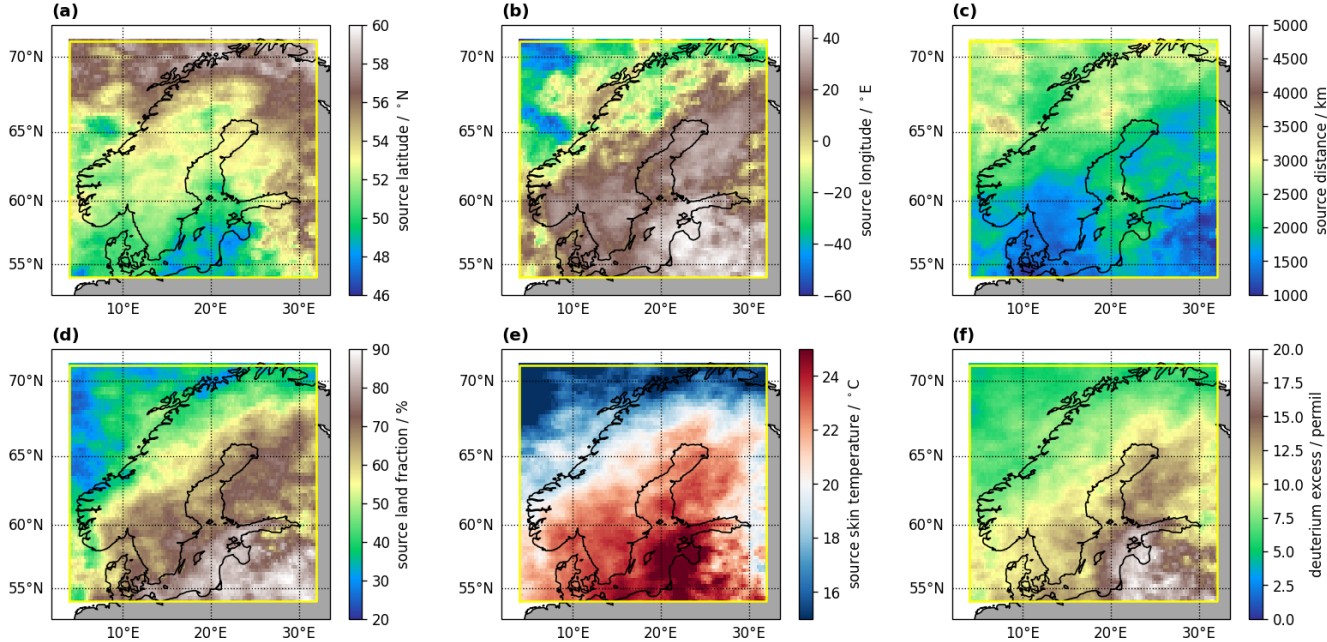

**Figure 6.** Lagrangian forward projection of source quantities for the example case in 2022. (a) Moisture source latitude (shading, °E), (b) moisture source longitude (shading, °N), (c) moisture source distance (shading, km), (d) moisture source land fraction (shading, %), (e) moisture source skin temperature (shading, °C), (f) d-excess of water vapour at the moisture source following Pfahl and Sodemann (2014) (shading, ‰). All quantities are weighted averages, using the Lagrangian precipitation estimate $\tilde{P}$ as a weight.

an air parcel trajectory. The difference plot between both distance quantities underlines that the transport distance is always larger than the source distance, and more so for regions where transport takes more time, with a differences of up to 2000 km in Northern Scandinavia (Fig. 7f). In combination, these transport quantities enable one to interpret the transported moisture in terms of processes that occurred underway from source to sink. Such information can for example be relevant for understanding changes of isotope composition and the mixing with other air masses. For more detailed discussion of the transport time from WaterSip, see Sodemann (2020) and Gimeno et al. (2021).

## 4.6 Time series of diagnostic quantities

The temporal evolution of different diagnostic quantities in the arrival domain can provide further insight into moisture transport dynamics. WaterSip provides time series output of all arrival and forward-projected source ($\zeta$) and transport quantities ($\xi$) described above (Sec. 3.3). Thereby, the mean, median, and the maximum and minimum value are reported for different integration times, from the trajectory time step to daily, monthly, yearly, and the entire analysis time period (Table 2). Averaging over time steps is done by weighting the contribution at each time step $i$ of the analysis period with the corresponding





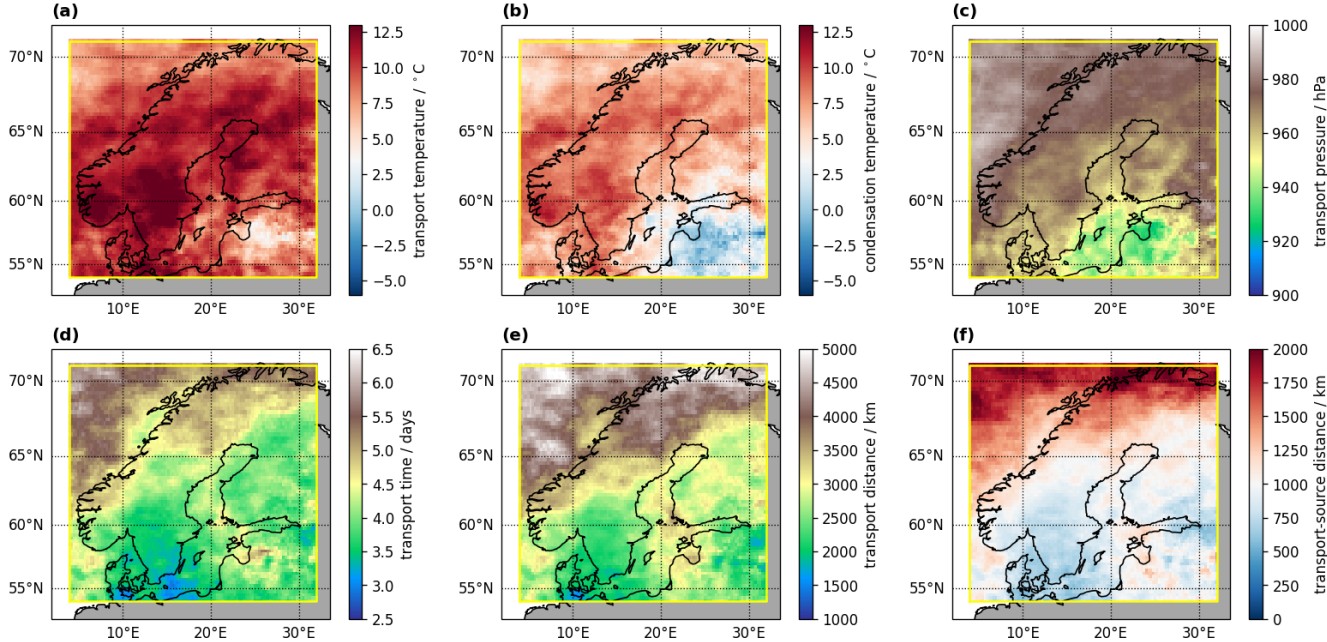

**Figure 7.** Lagrangian forward projection of transport quantities for the example case in 2022. (a) transport temperature (shading, °C), (b) condensation temperature (shading, °C), (c) transport pressure (shading, hPa), (d) transport time (shading, days), (e) transport distance (km). All quantities are weighted averages, using the Lagrangian precipitation estimate $\tilde{P}$ as a weight.

precipitation estimate $\tilde{P}$ at grid location $(\phi_a, \theta_a)$, and dividing by the total of the precipitation estimate for a given averaging period $T$:

$$\overline{\zeta}(\phi_a, \theta_a) = \frac{\sum_{i=0}^{T} \zeta_i(\phi_a, \theta_a) \cdot \tilde{P}_i(\phi_a, \theta_a)}{\sum_{i=0}^{T} \tilde{P}_i(\phi_a, \theta_a)} \tag{23}$$

Time series of $\tilde{P}$ of the case study show a pronounced precipitation maximum in the second half of the study period on 16 to 17 Aug 2022 with 1.3 mm $(6\,\mathrm{h})^{-1}$ (Fig. 8a). The period with most intense precipitation was associated with gradually shorter

transport distances of less than 3000 km (Fig. 8b). Land sources where dominating in this second period, with a fraction up to 75% on 17 Aug 2022, compared to below 30% on 12 Aug 2022 (Fig. 8c). Source latitudes varied more strongly during the first half of the case study, and stabilised at around 52°N after 15 Aug 2022 (Fig. 8d).

During the forward projection, some grid points have only a very small number of trajectories arriving, which can then lead to relatively large grid maximum and minimum values compared to the overall spread. For some applications, a more refined

statistical quantification of the gridded fields may be required, for example such as excluding grid locations below a minimal





precipitation estimate. In such a case, users may chose to output the gridded fields for every time step, and then follow up with their own post-processing steps.

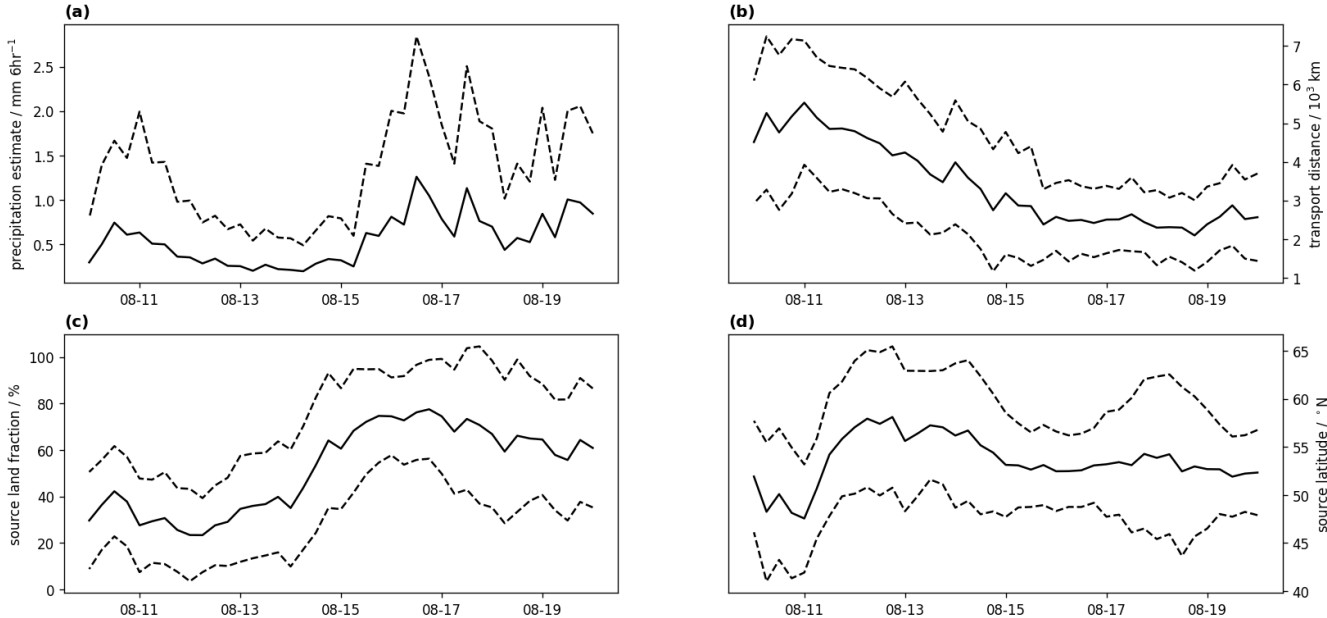

**Figure 8.** Time series output from WaterSip for the Scandinavian precipitation event in August 2022 for the variables (a) precipitation estimate (mm $6\,\mathrm{h}^{-1}$), (b) transport distance ($10^3$ km), (c) source land fraction (%), (d) source latitude (°N). Each panel shows the weighted mean (solid line), unweighted one-$\sigma$ standard deviation (dashed lines) over the arrival grid.

## 4.7 Sectorisation of source region maps

Another post-processing available within WaterSip is the option to categorise the moisture source maps by geographic domains, the so-called sectorisation (Sodemann and Zubler, 2010). During sectorisation, the coordinates of each uptake in the boundary layer and free troposphere are checked to be within a set of $J$ latitude and longitude bounds $[\lambda_j, \theta_j]$, which define the source sectors $S_j$. The geographic bounds of each sector are implemented in the program code as a function that returns the sector number corresponding to a given geographic coordinate. Currently, 11 different sectorisations are available in WaterSip (Table B1). Implementing additional sectors requires users to modify the WaterSip code. Further details are given in the 665 Appendix B.

The output from the sectorisation is included in the time series files. All quantities available on the source and transport grids contain an additional dimension for the sector number. Furthermore, quantities are given for land regions only, and for the total. This allows to define relatively simple geographic sector shapes, that still make the important distinction between land



and ocean regions, based on the land-surface information provided to WaterSip. The sector areas are also available to convert the fractional output to mass units.

For the case study over Scandinavia, the sectorisation 'Norway' has been chosen, which includes 8 sectors (Fig. 9c). The total arriving precipitation is dominated by sources at 30–40°N (Fig. 9a, light green) in the first half of the study period, before being dominated by more northerly sources in the second half (40–50°N, dark green). Splitting up the moisture originating from land areas only further emphasises that the shift to more northerly sources is driven by land contributions (Fig. 9c).

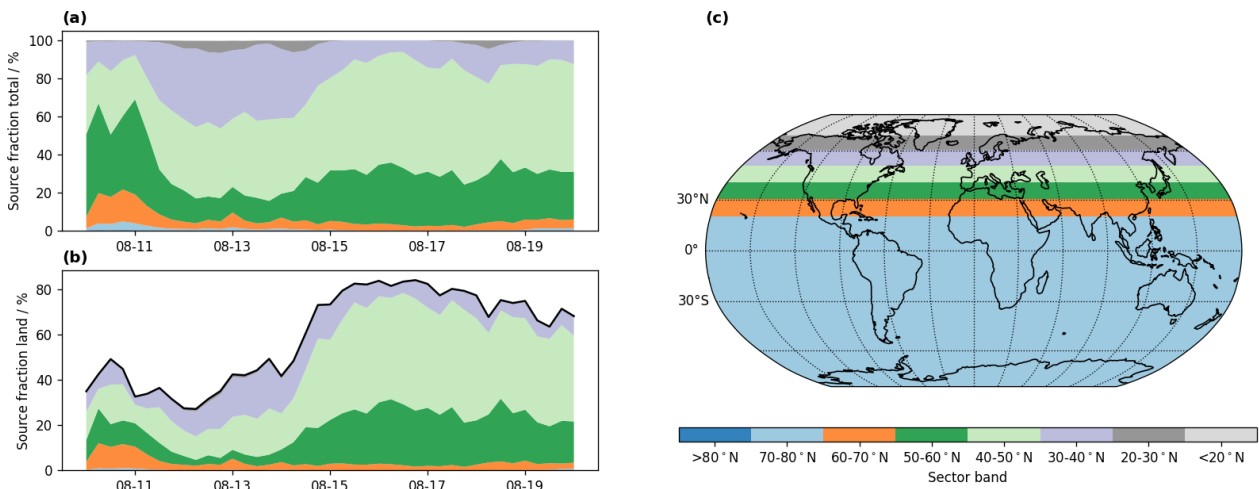

**Figure 9.** Sectorisation example for the Scandinavian case study for August 2022. (a) Classification of the total (land and ocean) moisture sources into latitude bands between 20–80°N during the study period. (b) Classification of the land sources only. Solid black line indicates the total fraction of all land sources. (c) Sector map of sectorisation 'Norway' (#3) used in the case study.

## 4.8 Histograms of moisture source and transport properties

For some applications, it may be valuable to have the source and transport information from WaterSip available from every single uptake event. During the gridding to source or arrival grids, this detail of information is lost. Since the number of individual uptakes can be very large, histograms are a convenient way to aggregate such data. For example, Winschall et al. (2014b) used histograms of transport time to identify the age spectrum of precipitation. Other examples for the use of such output are the distribution of SSTs at the moisture sources (Weng et al., 2021). WaterSip creates an output file in comma separated value (csv) format containing histograms of a fixed set of variables as output if the parameter *saveHistogram* in section *Output* is set to 1 (true). These variables include convection count and depth, uptakes, rainout, air mass mixing, source SST or T2m, source distance, source d-excess, and $RH_{SST}$ at the sources (Table 3). As the histogram classification is implemented directly within WaterSip, changes to the classification parameters, or adding more variables requires changes to the WaterSip program code in file `Water.cpp`.





For the example case of Scandinavia during August 2022, the probability density function of evaporation contribution shows a maximum 2 days before arrival, and a shallow tail beyond 10 days before arrival (Fig. 10a). The distribution of rainout events peaks at the same time, but has otherwise an even distribution throughout the 20 day analysis period (Fig. 10b). The PDF of mixing events peaks 5 days before arrival (Fig. 10c). Finally, the PDF of source distances shows a very flat tail beyond about 690 9000 km, with the majority of sources within about 3000 km (Fig. 10d). These examples illustrate the additional insight that one can gain from the histogram data, beyond the mass-weighted averages provided by the gridded and time series output.

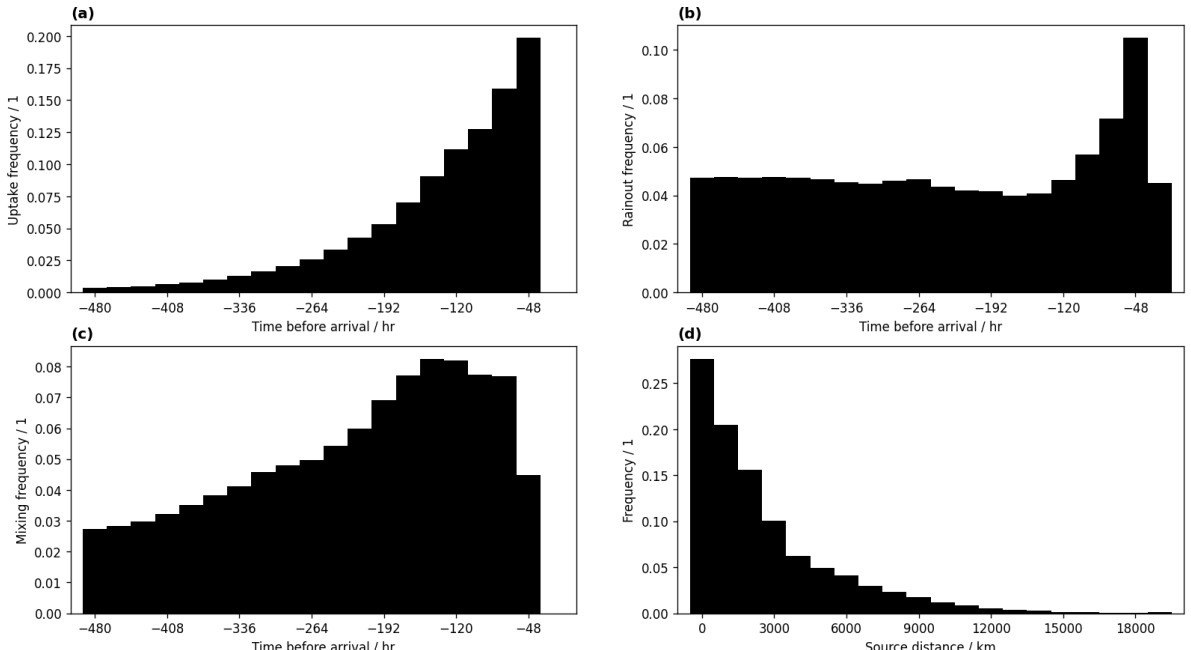

**Figure 10.** Histograms of moisture source and transport quantities for the example case. (a) Uptake frequency by time before arrival, (b) Rainout frequency by time before arrival, (c) Mixing frequency by time before arrival, and (d) source distance frequency by time before arrival. All histograms are valid for the entire analysis period and have been converted to probability density functions.

## 4.9 Air parcel trajectory output

WaterSip processes air parcel trajectory data obtained from the Lagrangian models LAGRANTO and FLEXPART. During the processing of these data, trajectories are filtered based on criteria such as being located within the arrival domain, precipitating 695 during arrival, and so on. Essentially, thus, the WaterSip tool creates a subset of the original trajectory data, enhanced by additional information, such as the identified uptakes, and the accounted fraction. Setting parameter *saveTraj* of settings group *Output* to either 1 (output every time step) or 2 (output in monthly files) instructs WaterSip to write out these trajectory data



**Table 3.** Histogram variables currently built into WaterSip, with corresponding lower and upper bounds, and number of categories for the classification.

| Parameter | Convection height (m) | Convection count (1) | Uptake (g kg$^{-1}$) | Uptake contribution (g kg$^{-1}$) | Rainout (g kg$^{-1}$) | Air mass mixing (g kg$^{-1}$) | Source distance (km) | SST (K) | d-excess (permil) | RH$_{SST}$ (%) |
|---|---|---|---|---|---|---|---|---|---|---|
| min | 0 | 0 | -480 | -480 | -480 | -480 | 0 | 270 | -10 | 0 |
| step | 10 | 10 | 20 | 20 | 20 | 20 | 20 | 20 | 20 | 20 |
| max | 2000 | 10 | 0 | 0 | 0 | 0 | 20000 | 310 | 30 | 100 |

in a compact binary format. Since the total output size can become rather large, it is possible to use parameter *skipTraj* with a value $n > 1$ to only write out every $n$-th trajectory.

The potential value of the trajectory output is illustrated here for a subset of the air masses arriving over the target area in Scandinavia at 06 UTC on 10 Aug 2022. During this time, 20-day backward trajectories tap into moist air masses near the equator in the Atlantic and Pacific basin (Fig. 11a). The colour shading by moisture uptakes and losses emphasises regions of moisture loss over the Rocky Mountains (red), moisture gain over eastern North America (blue), and as air parcel trajectories arrive over Scandinavia (red). Plotting the total accounted fraction in a time vs height diagram emphasises that moisture uptakes

of more than 10 days before arrival are mostly overwritten by later uptake events (Fig. 11b).

## 5    Discussion

As mentioned throughout the description of the method, there are a number of uncertainties connected to the use of the WaterSip diagnostic. Users need to carefully consider the impact such uncertainties may have on the interpretation and validity of their results (Sec. 5.1). Underlying this uncertainty is the lack of a direct observable quantity that would serve as a reference for

finding out moisture source information. In that context, method inter-comparisons and sensitivity studies can be a pathway to quantify and potentially bound uncertainty (Sec. 5.2).

### 5.1    Error sources and uncertainties

The WaterSip algorithm, as other Lagrangian diagnostics, calculates the Lagrangian water budget of air parcels that are moving with the atmospheric flow. At initial time $t_0$, these air parcels are defined by their location, spatial extent, and the air parcel

with mass $m$. As the air parcels are moving horizontally and vertically, the shape is assumed to be maintained, as well as the total mass. However, exchange processes across the air parcel's imaginary walls can modify the amount of water vapour within the air parcel, as quantified by the specific humidity $q$ (g kg$^{-1}$). Interpolation errors can then, among other, cause spurious variations in the specific humidity. While threshold parameters, such as $\Delta q_c$ are used to suppress some of the numerical noise introduced during the trajectory calculation, there may either be to many uptakes diagnosed, or some uptake events may be

missed. The total explained fraction $f_{tot}$ can then provide important guidance whether the accounting works consistently (i.e.,



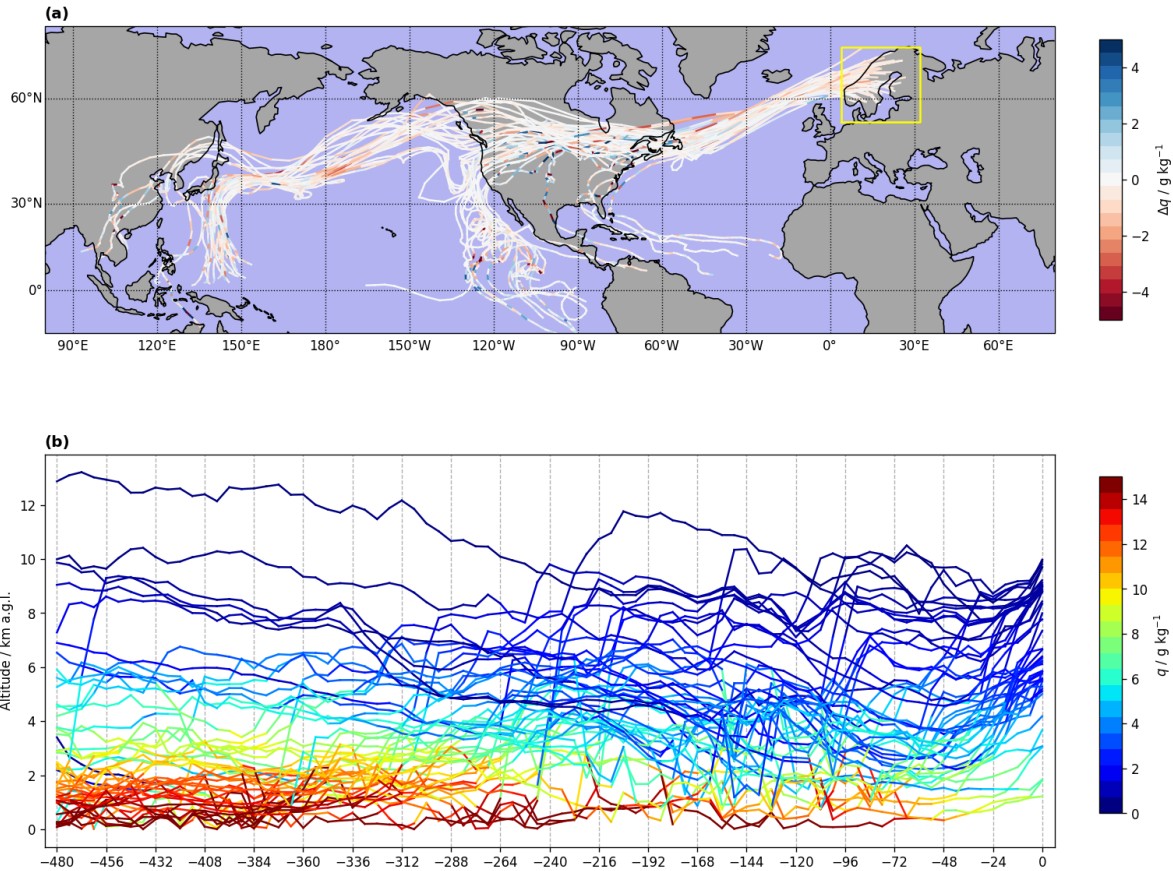

**Figure 11.** Trajectory output from WaterSip for 06 UTC on 10 Aug 2022. (a) Map of the moisture transport pathway of 20-day trajectories shaded by specific humidity changes per time step ($\Delta q$ in g kg-1 6h$^{-1}$). (b) Time vs. height diagram shaded by specific humidity (g kg$^{-1}$) for water vapour arriving in the Scandinavia domain. For illustrative purposes, a random selection of 50 out of more than 1000 air parcel trajectories is displayed.

$f_{\text{tot}} \leq 1$) with the chosen threshold parameters. Spurious variations that are larger than the threshold value will lead to more remote moisture being discounted in the favour of later uptakes. Such potential biases are important to keep in mind during interpretation.

Other important sources of uncertainty are the number of particles chosen to represent the investigated air mass, as well as the time step of the trajectory output. In general, a larger number of trajectories will provide better statistical representation of the diagnosed quantities, in particular when used with a particle dispersion model such as FLEXPART that parameterises turbulent and convective motions. The number of trajectories should also be chosen in line with the resolution of the source and arrival grids, providing reasonably smooth and continuous gridded output. Other considerations of offline trajectory diagnostics



are the effects of data assimilation, which provides humidity increments during the process (Fremme et al., 2023), and from parameterisation schemes in the parent numerical model. Here in particular the role of convection schemes is important to consider, providing both a pathway for rapid specific humidity changes and vertical particle motion that are hard to capture in offline calculations, even when using parameterisation schemes for convection in the trajectory model (Sodemann, 2020). Here, the use of online trajectory calculation methods (Miltenberger et al., 2013) could provide a pathway towards quantifying such uncertainty.

## 5.2 Quantifying and reducing uncertainty

Future development of the WaterSip software and similar approaches will benefit from quantifying uncertainty, and from identifying the contribution of different factors to uncertainty. In the absence of a direct observable quantity for the water vapour origin, model inter-comparison efforts are important. First steps in this direction have been made by van der Ent et al. (2013) who employed a regional tagged water tracer simulation as the reference standard for a Eulerian moisture source identification algorithm. Similarly, Winschall et al. (2014a) and Laederach (2016) compared the WaterSip diagnostic to tagged tracer simulations from a regional model. While tagged tracers do have limitations with regard to the representation of different processes in the atmospheric water cycle at small scales, it can be expected that using such online simulations as input data for offline diagnostics can help to quantify uncertainty, essentially serving as an internal "gold standard" for such inter-comparison efforts (Sodemann, 2020). Thanks to the availability of an increasing number of approaches to provide (Lagrangian and Eulerian) offline moisture source diagnostics as open access software (van der Ent et al., 2014; Keune et al., 2022; Fernández-Alvarez et al., 2022), the feasibility of such an inter-comparison effort has greatly increased in recent years. The present description of the WaterSip diagnostic is another piece in the puzzle that may help to advance ongoing debates. On the longer run, the availability of suitable water vapour isotope measurement data sets, in particular such that connect moisture source signals with measurements along the transport pathway and the arrival region, have the potential to indeed constrain diagnostic methods by measurements (Gimeno et al., 2021).

## 6 Summary and Conclusions

The WaterSip software is a versatile implementation of the Lagrangian diagnostics of moisture sources and transport diagnostic of Sodemann et al. (2008b). A detailed, formal description of the diagnostic algorithm and its key parameters will be useful in inter-comparison studies with other tools for moisture source identification, supporting the understanding of any differences between methods. Convenient data aggregation in various ways allows researchers to focus on the results and interpretation. The format and contents of the different output files from the diagnostic are described. Furthermore, set-up for trajectory calculations and the parameters of the moisture source diagnostic are discussed. Using a case study for a precipitation period over Scandinavia in August 2022, guidance on the interpretation of different maps and output products from the software is provided. Using the available setup with the same input data as the case study (Sodemann, 2025), users can get quickly



accustomed with the wide range of configuration options of the software tool, and the varieties of diagnostic output, before setting up their own cases.

Availability of the software tool enables a wider use of this diagnostic for studies of the moisture source location and properties, as well as for identifying the conditions during atmospheric transport, and during arrival target region. Making the software code available in a public repository removes the requirement for other researchers to re-implement the diagnostic

algorithm, and enables future contributions from the community such as reading output from other Lagrangian models than LAGRANTO and FLEXPART. This manuscript provides essential guidance on how the algorithm works, how to set up the diagnostic, and recommends best practices for interpreting the results, and for working with uncertainties.

Being the implementation of an off-line diagnostic with several underlying assumptions, interpreting results from WaterSip necessitates the careful consideration of uncertainty. Uncertainty arises from the choice of different parameters in the set-up

of the input data in the form of air parcel transport trajectories, but also with regard to several parameters that influence the sensitivity and performance of the method. In order to obtain physically meaningful results, threshold parameters may need to be set to different values depending on the geographic region, the season, and the height in the atmosphere that is considered during an analysis. Users are therefore advised to carefully test their particular set-up in order to determine sensitivity of their results to the chosen parameter configurations.

It is the hope of the author that documenting and making the code of WaterSip available here will lead to the further application of the tool in a wide range of regions and scientific applications, from understanding the factors of precipitation extremes, to climatological studies, and paleoclimate studies. Furthermore, WaterSip in its current version 3.2 may be compared to the results of other available moisture source diagnostics, helping to understand the overall state of the art in this scientific branch, thereby contributing to progress in the field and future methods with reduced uncertainty.

*Code and data availability.* The model code of the WaterSip diagnostic in its current version 3.2 is available at the git repository https://gitlab.uib.no/gfi-public/watersip. Test data sets with FLEXPART output and corresponding WaterSip results are available at https://doi.org/10.5281/zenodo.14836549 (Sodemann, 2025).

## Appendix A: Description of control parameters in the WaterSip configuration file

The WaterSip configuration file contains all control parameters of the WaterSip diagnostic in one text file. The control pa-
rameters are each specified on one line in the format ` = `. Parameters can be either a string, integer, boolean, or float value. Parameters are grouped into one of 7 settings groups. Corresponding tables referred to in the Appendix provide an overview over all parameters in this group, including example values.

1. Case: The settings in this group concern the basic settings of a diagnostic run (Table A1).

2. Grids: The parameters in this category specify the different output grids and the gridding method (Table A2).



3. Diagnostics: The parameters in this category control the way how the diagnostic works, and which trajectories should be considered in the run (Table A3).

4. Output: The parameters in this category determine which output files should be created (Table A4).

5. Variables: The parameters in this category allow to include or exclude specific variables from the output files. Setting the output for unused variables to 0 (disabled) will speed up computation time, reduce the memory footprint, and reduce the

output file size (Table A5).

6. FLEXPART: The parameters in this category describe the properties specific to FLEXPART input files (Table A6).

7. LAGRANTO: The parameters in this category describe the properties specific to LAGRANTO input files (Table A7).

Settings groups are identified in the configuration file by their name included within in square brackets (e.g. `[Case]`). The 7 settings groups and control parameters within them can appear in random order in the input file. WaterSip will display an

error message if a parameter name is unknown or in the wrong settings group. An error message will also be displayed if the parameter value is of wrong type.

### Appendix B: Sectorisation options

The sectorisation parameters allow to assign quantities on the moisture source and transport grid to pre-specified regions (Sec. 4.7). A total of 14 sectorisations is currently built into the program code of WaterSip, covering different regions of the

globe (Fig. B1, Table B1). The sectorisation is specified in the run setup using parameter *sectorizeRegion* in settings group *Case*. Each of the sectors distinguishes between the respective land and ocean area. During post-processing, it is common to combine several sectors together to obtain desired information over a source region with more complex shape than a box. Users can define new sectorisations by modifying the program code in class Sectors (files `Sectors.cpp, Sectors.h`).

### Appendix C: Output files

WaterSip creates a range of output files with different formats (Sec. 3.3 and Table 2). The primary results from the diagnostic are contained in the grid and series files. While grid files contain the spatially resolved information for different averaging intervals (time step, daily, monthly, yearly, and whole analysis period), the series files contain statistical properties (sum or mean, standard deviation, minimum and maximum) per averaging time. The source and transport properties are contained in grid and series files (Table C1), together with arrival and forward-projected quantities (Table C2).

The series file furthermore contains the results of the sectorisation (Sec. 4.7). Sectorisation variables comprise the sector area, as well as a percent fraction of the source and transport properties available on the global map per sector (Table C3). The sector information is separated into land and sea fractions, whereby the sector number is can be accessed along the dimension *sector*.



**Table A1.** Parameters in the settings group Case.

| Parameter | Type | Example | Description |
|---|---|---|---|
| caseName | s | test_global | identifier for the run, prepended to all output files |
| inputDir | s | ./input/shortposit_%s | Search path of the input files. The inputDir string ends with a placeholder for the files to be read in. For FLEXPART this will be either shortposit_%s or partposit_%s. For LAGRANTO this will often be lsl_%s. The %s placeholder is filled with the current date at runtime. |
| outputDir | s | ./output/test_global/ | path where the output files will be created |
| startDate | s | 20230811-000000 | Inital date of the run, fixed format YYYYMMDD-HHMMSS. Start date is latest date of the run. |
| endDate | s | 20230713-030000 | Last date of the run, fixed format YYYYMMDD-HHMMSS. End date is earliest date of the run. |
| timeStep | i | 3 | Time step in hours between particles or trajectory output times. |
| sectorizeRegion | i | 6 | Region set used to divide moisture sources into different sectors. |
| filterBoxFile | s | " " | Path to the text file describing geographic boxes used to filter air parcels during transport. |
| filterIndex | i | -1 | Used line from the filterBoxFile to select/deselect trajectories |
| orographyFile | s | ./topo/etopo5.nc | Path to the orography file in netCDF format used to select arrival based on the underlying topography. |
| lsmFile | s | ./masks/lsm_mask.nc | Path to a netCDF file with global land-sea-mask. Domain must be global. |
| maskFile | s | " " | Path to a netCDF file with an arrival region mask. |
| useMask | b | 0 | Use mask file (0=no, 1=yes) |
| trajPoints | i | 161 | Number of trajectory points to be considered in the analysis of arriving air parcels. |
| minTrajPoints | i | 0 | Minimum length required for trajectories to be included in analysis (only relevant for runs with LAGRANTO input). |
| ompThreads | i | 4 | Number of threads to be used with OpenMP parallelisation if compiled with flag -fopenmp (1=no parallelisation, $>$1 use parallelisation.) |
| showStats | b | 1 | Print statistics about particle processing to the command line (0=off, 1=on) |

*Author contributions.* HS wrote the manuscript, wrote and tested the software code, and performed visualisation and data analysis.

*Competing interests.* The author declares no competing interests

*Acknowledgements.* I would like to acknowledge the many colleagues who contributed through their interest and comments, curiosity and critical questions to the further development of the WaterSip tool, in particular Alexander Läderach, Astrid Fremme, Yongbiao Weng, Mika Lanzky, and Matthew Osman. Marte Hofsteenge and Costijn Zwart are kindly acknowledged for providing helpful comments to an earlier version of this manuscript.



**Table A2.** Parameters in the settings group Grid.

| Parameter | Type | Example | Description |
|---|---|---|---|
| arrivalGridMinLon | f | 7.5 | Minimum longitude of arrival (target region) grid. Air parcel trajectories are traced backward from the arrival grid. Integrated source, transport and arrival variables are placed on the arrival grid, weighted by precipitation or water vapour of arriving air parcels. |
| arrivalGridMaxLon | f | 11.0 | maximum longitude of arrival (target region) box |
| arrivalGridMinLat | f | 59.0 | minimum latitude of arrival (target region) box |
| arrivalGridMaxLat | f | 62.0 | minimum latitude of arrival (target region) box |
| arrivalGridDx | f | 0.25 | Longitude increment of arrival (target region) grid. |
| arrivalGridDy | f | 0.25 | Latitude increment of arrival (target region) grid. |
| arrivalGridRadius | f | 40 | Gridding radius for diagnostic values from arriving air parcels. |
| sourceGridMinLon | f | -120 | Minimum longitude of moisture source grid. |
| sourceGridMaxLon | f | 120 | Maximum longitude of moisture source grid. |
| sourceGridMinLat | f | 0 | Minimum latitude of moisture source grid. |
| sourceGridMaxLat | f | 90 | Maximum latitude of moisture source grid. |
| sourceGridDx | f | 0.5 | Longitude increment of moisture source grid. |
| sourceGridDy | f | 0.5 | Latitude increment of moisture source grid. |
| sourceGridRadius | f | 60 | Longitude increment of moisture source grid. |
| griddingType | i | 2 | Select gridding algorithm (1: fast and simple gridding, 2: more exact but slower gridding). |

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



**Table A3.** Parameters in the settings group Diagnostics.

| Parameter | Type | Example | Description |
|---|---|---|---|
| uptakeThreshold | f | 0.2 | Threshold value for specific humidity increase (in g kg$^{-1}\Delta t^{-1}$) in air parcels to be considered as due to evaporation and transpiration. If the parameter reativeThresholds is set to true, the uptakeThreshold specifies the increase of the specific humidity in units of %. |
| precipThreshold | f | -0.2 | Threshold value for specific humidity decreases (in g kg$^{-1}\Delta t^{-1}$) in air parcels to find precipitation events. When relativeThresholds is set to true, the precipThreshold specifies the decrease of the specific humidity in %. For consistency, it is recommended to set the values for precipitationThreshold and arrivalPrecipMin to the same number. |
| blhScale | f | 1.5 | Scaling factor for boundary-layer height, decides wether uptakes are due to surface evaporation from below, or due to free-troposphere processes, such as convection and mixing. |
| arrivalRHMin | f | 80 | Threshold value for minimum RH (in %) at the arrival location that identifies moisture losses as precipitation events. High latitude locations may require lower threshold values. |
| arrivalRHMax | f | 120 | Threshold value for maximum RH (in %) at the arrival location that identifies moisture losses as precipitation events. In combination with arrivalRHMin, this parameter can be used to focus on condensation/no condensation cases when the parameter analyzeVapour is set to true. |
| arrivalPrecipMin | f | -0.2 | Threshold value to find precipitation events from a specific humidity decrease (in g kg$^{-1}\Delta t^{-1}$). When relativeThresholds is set to true, the precipThreshold specifies the decrease of the specific humidity in %. For consistency, it is recommended to set the values for precipitationThreshold and arrivalPrecipMin to the same number. |
| arrivalPrecipMax | f | -120 | Threshold value to consider precipitation events with a specific humidity decrease of up to the value given by arrivalprecipMax. When relativeThresholds is set to true, the precipThreshold specifies the decrease of the specific humidity in %. A large negative value includes all potential precipitation events. The combination of arrivalPrecipMin and arrivalPrecipMax allows to focus on heavy or weak precipitation events. |
| arrivalOroMin | f | -100 | Threshold value to consider only arrival locations with a topography larger than the given value (in m). Use negative value to include all global locations including ocean areas, Bangladesh and Death Valley. |
| arrivalOroMax | f | 90000 | Threshold value to consider only arrival locations with a topography lower than the given value (in m). |
| arrivalAltMin | f | -100 | Threshold value to consider only air parcels arriving at altitudes above the given altitude (in m above sea level). |
| arrivalAltMax | f | 90000 | Threshold value to consider only air parcels arriving at altitudes below the given altitude (in m above sea level). |
| analyzeVapour | b | 0 | Parameter to chose between precipitation origin (0) or vapour origin (1). When analyzing vapour origin, the parameters precipitationThreshold, arrivalRHMin, and arrivalRHMax may need modification. |
| assignToUptake | b | 0 | Parameter determining the assignment of moisture uptake events to arrival time (0) or to uptake time step grid (1). This parameter will mostly affect time step and daily output files. |
| forwardProjectionMode | i | 3 | Parameter determining how the Lagrangian forward projection of moisture source and moisture transport properties are calculated (1: only consider boundary-layer moisture uptakes, 2: only consider free-troposphere uptakes, 3: consider both uptakes in combination). |
| relativeThreshold | b | 0 | Parameter determines whether specified threshold values are interpreted as absolute (0) or relative values (1). |



**Table A4.** Parameters in the settings group Output.

| Parameter | Type | Example | Description |
| --- | --- | --- | --- |
| stepFile | b | 0 | Write output variables on Grid files for each time step. |
| dayFile | b | 1 | Write output variables on grid files averaged/accumulated for each day. |
| monthFile | b | 1 | Write output variables on grid files averaged/accumulated for each month. |
| yearFile | b | 1 | Write output variables on grid files averaged/accumulated for each year. |
| allFile | b | 1 | Write output variables on grid files averaged/accumulated for entire analysis period. |
| staticFields | b | 1 | Write static fields (land-sea mask and topography) to grid files. |
| saveTraj | i | 0 | Write particle positions to trajectory files in traj format. Option (1) creates one files for each time step, option (2) one file for each month (2). Option (0) disables trajectory output. |
| skipTraj | i | 1 | Only write every n-th trajectory to reduce trajectory file size. Option (1) writes all particle trajectories. |
| saveHistogram | b | 1 | Write a subset of variables to histogram files (0: disabled, 1:enabled). |

Bonne, J. L., Masson-Delmotte, V., Cattani, O., Delmotte, M., Risi, C., Sodemann, H., and Steen-Larsen, H. C.: The isotopic composition of water vapour and precipitation in Ivittuut, southern Greenland, Atmos. Chem. Phys., 14, 4419—4439, https://doi.org/10.5194/acp-14-4419-2014, 2014.

Bonne, J.-L., Steen-Larsen, H. C., Risi, C., Werner, M., Sodemann, H., Lacour, J.-L., Fettweis, X., Cesana, G., Delmotte, M., Cattani, O., Vallelonga, P., Kjær, H. A., Clerbaux, C., Sveinbjörnsdóttir, A. E., and Masson-Delmotte, V.: The summer 2012 Greenland heat wave: In situ and remote sensing observations of water vapor isotopic composition during an atmospheric river event, J. Geophys. Res., 120, 2970–2989, https://doi.org/10.1002/2014JD022602, 2015.

Brubaker, K. L., Entekhabi, D., and Eagleson, P. S.: Estimation of continental precipitation recycling, J. Climate, 6, 1077–1089,
https://doi.org/10.1175/1520-0442(1993)006<1077:EOCPR>2.0.CO;2, 1993.

Buizert, C., Sigl, M., Severi, M., Markle, B., Wettstein, J., McConnell, J., Pedro, J., Sodemann, H., Goto-Azuma, K., Kawamura, K., Fujita, S., Motoyama, H., Hirabayashi, M., Uemura, R., Stenni, B., Parrenin, F., He, F., Fudge, T., and Steig, E. J.: Abrupt ice-age shifts in southern westerly winds and Antarctic climate forced from the north, Nature, 563, 681–685, 2018.

Dirmeyer, P. A. and Brubaker, K. L.: Contrasting evaporative moisture sources during the drought of 1988 and the flood of 1993, J. Geophys.
Res., 104, 19 383–19 397, http://dx.doi.org/10.1029/1999JD900222, 1999.

Dütsch, M., Pfahl, S., Meyer, M., and Wernli, H.: Lagrangian process attribution of isotopic variations in near-surface water vapour in a 30-year regional climate simulation over Europe, Atmospheric Chemistry and Physics, 18, 1653–1669, https://doi.org/10.5194/acp-18-1653-2018, 2018.

Eltahir, E. A. B. and Bras, R. L.: Precipitation recycling in the Amazon basin, Q. J. Roy. Meteor. Soc., 120, 861–880, http://dx.doi.org/10.
1002/qj.49712051806, 1994.

Eltahir, E. A. B. and Bras, R. L.: Precipitation recycling, Reviews of Geophysics, 34, 367–378, https://doi.org/https://doi.org/10.1029/96RG01927, 1996.

Fernández-Alvarez, J. C., Pérez-Alarcón, A., Nieto, R., and Gimeno, L.: TROVA: TRansport Of water VApor, SoftwareX, 20, 101 228, https://doi.org/https://doi.org/10.1016/j.softx.2022.101228, 2022.



**Table A5.** Parameters in the settings group Variables. Output of variables activated by parameter value (1) and deactivated by (0). Variables are written to grid and series output files.

| Parameter | Type | Description |
|---|---|---|
| sourcesBoundaryLayer | b | Write accounted moisture uptakes in the boundary layer to output files. |
| sourcesFreeTroposphere | b | Write accounted moisture uptakes in the free troposphere to output files. |
| evaporationMinusPrecipitation | b | Write unweighted evaporation minus precipitation along entire trajectory length to output files. |
| moistureTransport | b | Write weighted moisture transport along trajectories to output files. |
| trajectoryLocations | b | Write trajectory location density to output files. |
| precipitationEstimate | b | Write precipitation estimate at arrival to output files. |
| sourceLongitude | b | Write weighted moisture source longitude projected to the arrival region to output files. |
| sourceLatitude | b | Write moisture source latitude projected to the arrival region to output files. |
| arrivalCount | b | Write number of arriving trajectories on the arrival region to output files. |
| landFraction | b | Write fraction of moisture sources over land projected to the arrival region to output files. |
| boundaryLayerFraction | b | Write accounted fraction of boundary layer uptakes projected to the arrival region to output files. |
| freeTroposphereFraction | b | Write accounted fraction of free troposphere uptakes projected to the arrival region to output files. |
| combinedFraction | b | Write accounted fraction of boundary layer and free troposphere uptakes projected to the arrival region to output files. This is the sum of the two previous variables. |
| transportTime | b | Write moisture transport time projected to the arrival region to output files. |
| sourceSkinTemperature | b | Write skin temperature at the moisture source projected to the arrival region to output files. Requires that skin temperature is a variable on the input files. |
| sourceSpecificHumidity | b | Write surface specific humidity at the moisture source projected to the arrival region to output files. Requires that source specific humidity is a variable on the input files. |
| transportDistance | b | Write transport distance projected to the arrival region to output files. |
| sourceDistance | b | Write source distance projected to the arrival region to output files. |
| source2mTemperature | b | Write 2m temperature at the moisture source projected to the arrival region to output files. Requires that 2m temperature is a variable on the input files. |
| sourceDeuteriumExcess | b | Write deuterium excess projected to the arrival region to output files. Deuterium excess is calculated from the relative humidity with respect to SST and the relationship from Pfahl and Sodemann (2014). Requires that source specific humidity and SST or 2m temperature and humidity are variables on the input files. |
| transportTemperature | b | Write moisture-weighted temperature during transport projected to arrival region to output files. |
| transportPressure | b | Write moisture-weighted pressure during transport projected to arrival region to output files. |
| condensationTemperature | b | Write moisture-weighted temperatures where RH with respect to liquid exceeds 80% to output files. |
| arrivalTemperature | b | Write temperature during moisture arrival to output files. |
| arrivalPressure | b | Write pressure during moisture arrival to output files. |
| arrival2mTemperature | b | Write temperature at 2m during moisture arrival to output files. Requires that 2m temperature is a variable on the input files. |
| arrivalSkinTemperature | b | Write skin temperature during moisture arrival to output files. Requires that skin temperature is a variable on the input files. |

Flatau, P. J., Walko, R. L., and Cotton, W. R.: Polynomial Fits to Saturation Vapor Pressure, Journal of Applied Meteorology and Climatology, 31, 1507 – 1513, https://doi.org/10.1175/1520-0450(1992)031<1507:PFTSVP>2.0.CO;2, 1992.





**Table A6.** Parameters in the settings groups FLEXPART.

| Parameter | Type | Example | Description |
|---|---|---|---|
| inputFormat | i | 5 | File format of FLEXPART particle position output files. 0: regional long output format, 1: global analysis data output format, 2: regional short format, 3: global reanalysis/NorESM data short format, 4: global reanalysis data long format, 5 regional reanalysis/NorESM data short format. Formats are implemented in the WaterSip code in file readparticles.cpp. |
| maxPart | i | 4000000 | Number of particles contained in a single FLEXPART partposit or shortposit file. |
| partMass | f | 2.54E+11 | Atmospheric mass represented by each particle (in kg). Parameter value corresponds to value in file "partmass" created by FLEXPART. |
| allowReenter | b | 0 | Needs to be enabled (1) for global domain-filling simulations. In regional domain-filling runs this parameter needs to be deactivated (0) to avoid double accounting of air parcels. |
| partStride | i | 1 | Particles to skip during processing to speed up calculations for testing (1: use every particle, n>1: use every n-th particle). |

**Table A7.** Parameters in the settings groups LAGRANTO. Variables are identified by column index, starting from 0. Variables not contained in trajectory files are specified as value -1.

| Parameter | Type | Example | Description |
|---|---|---|---|
| partMass | f | 4E+06 | Particle mass per arriving trajectory, obtained as $\Delta x \cdot \Delta y \cdot \Delta p$ in units of m$^2$ hPa. |
| inputFormat | i | 1 | Choice between different formats of the LAGRANTO file. Format where the header contains time in HH (hours) only set by (0), format with header time in HH:MM (hours and minutes set by (1). |
| indexPressure | i | 3 | Column index of variable air parcel pressure (in hPa). |
| indexBoundaryLayerHeight | i | 11 | Column index of variable boundary-layer height (in m). |
| indexSkinTemperature | i | 12 | Column index of variable surface skin temperature (in K). |
| indexPotentialTemperature | i | 6 | Column index of variable potential temperature (in K). |
| indexSpecificHumidity | i | 10 | Column index of variable specific humidity (in g kg$^{-1}$). |
| indexEvaporation | i | -1 | Column index of variable surface evaporation (in mm). |
| indexTemperature | i | 7 | Column index of variable air temperature (in K) |
| indexRelativeHumidity | i | 8 | Column index of variable relative humidity (in %) |
| indexHeight | i | -1 | Column index of variable height (in m a.g.l.) |
| indexZonalWind | i | -1 | Column index of variable zonal wind (in m s$^{-1}$) |
| indexMeridionalWind | i | -1 | Column index of variable meridional wind (in m s$^{-1}$) |
| indexIntegratedWaterVapour | i | -1 | Column index of variable integrated water vapour (in mm) |
| index2mTemperature | i | -1 | Column index of variable 2m air temperature (in K) |
| indexOrography | i | -1 | Column index of variable topography height (in m a.s.l.) |

Fremme, A. and Sodemann, H.: The role of land and ocean evaporation on the variability of precipitation in the Yangtze River valley, Hydrol. Earth Syst. Sci., pp. 2525–2540, https://doi.org/10.5194/hess-23-2525-2019, 2019.



**Table B1.** Sectorisations that are currently available in WaterSip by setting parameter *sectorizeRegion* in settings group *Case*.

| sectorizeRegion | Name | Number of sectors |
|---|---|---|
| 0 | NEEM | 11 |
| 1 | Antarctica | 21 |
| 2 | Borneo | 14 |
| 3 | Norway | 8 |
| 4 | Arctic | 21 |
| 5 | Belize | 15 |
| 6 | Alps | 9 |
| 7 | Pakistan | 6 |
| 8 | Latitude | 18 |
| 9 | Longitude | 36 |
| 10 | China | 19 |
| 11 | Scandinavia | 23 |
| 12 | Arctic Seas | 19 |
| 13 | Pakistan 2022 | 13 |

**Table C1.** Source and transport quantities contained in the grid and series files. Each variable is available as the sum, the standard deviation over the non-zero grid points, and the minimum and maximum value of the respective averaging interval. For step files, the precipitation rate is given per time step $\Delta t$, for all other files the rate is per day.

| Variable | Units |
|---|---|
| moisture_uptakes_boundary_layer_(sum\|std\|min\|max) | mm $(\Delta t$ or day$)^{-1}$ |
| moisture_uptakes_free_troposphere_(sum\|std\|min\|max) | mm $(\Delta t$ or day$)^{-1}$ |
| evaporation_minus_precipitation_(sum\|std\|min\|max) | mm $(\Delta t$ or day$)^{-1}$ |
| rainout_(sum\|std\|min\|max) | mm $(\Delta t$ or day$)^{-1}$ |
| air_mass_mixing_(sum\|std\|min\|max) | mm $(\Delta t$ or day$)^{-1}$ |
| moisture_transport_(sum\|std\|min\|max) | mm |
| trajectory_location_(sum\|std\|min\|max) | counts km$^{-2}$ |



**Table C2.** Arrival and Lagrangian forward projected properties contained in the grid files. $^*$ denotes variables weighted in addition by the Lagrangian precipitation estimate $\tilde{P}$. $^+$ denotes variables weighted by the accounted fraction.

| Name | Unit | Projection | Description |
|---|---|---|---|
| precipitation_estimate | mm day$^{-1}$ or mm $\Delta t^{-1}$ | arrival | Lagrangian precipitation estimate at arrival location, derived from change in specific humidity during last time step |
| arrival_temperature* | K | arrival | temperature at trajectory location and elevation during arrival |
| arrival_pressure* | hPa | arrival | pressure at trajectory location and elevation during arrival |
| arrival_skin_temperature* | K | arrival | SKT below trajectory location during arrival |
| accounted_fraction_boundary_layer | fraction | arrival | fractional contribution of BL moisture sources to $\tilde{P}$ |
| accounted_fraction_free_troposphere | fraction | arrival | fractional contribution of FT moisture sources to $\tilde{P}$ |
| precipitation_lifetime*+ | h | transport | precipitation life time |
| transport_distance*+ | km | transport | transport distance along trajectory |
| moisture_transport_temperature*+ | K | transport | temperature during moisture transport |
| moisture_transport_pressure*+ | hPa | transport | pressure during moisture transport |
| condensation_temperature*+ | K | transport | temperature when RH$>$ 80% during transport |
| source_longitude*+ | degrees_east | source | latitude of the moisture source |
| source_latitude*+ | degrees_north | source | longitude of the moisture source |
| land_fraction*+ | fraction | source | fraction of moisture sources over land |
| source_skin_temperature*+ | K | source | SKT at the moisture source |
| source_distance*+ | km | source | direct distance between moisture source and arrival location |
| deuterium_excess*+ | ‰ | source | estimate of deuterium excess at the source from RH and T using the Pfahl and Sodemann (2014) relation |

Fremme, A., Hezel, P. J., Seland, Ø., and Sodemann, H.: Model-simulated hydroclimate in the East Asian summer monsoon region
during past and future climate: a pilot study with a moisture source perspective, Weather and Climate Dynamics, 4, 449–470, https://doi.org/10.5194/wcd-4-449-2023, 2023.

Gimeno, L., Eiras-Barca, J., Durán-Quesada, A. M., Dominguez, F., van der Ent, R., Sodemann, H., Sánchez-Murillo, R., Nieto, R., and Kirchner, J. W.: The residence time of water vapour in the atmosphere, Nature Reviews Earth & Environment, 2, 558–569, https://doi.org/10.1038/s43017-021-00181-9, 2021.

Gustafsson, M., Rayner, D., and Chen, D.: Extreme rainfall events in southern Sweden: where does the moisture come from?, Tellus A, 62, 605–616, https://doi.org/https://doi.org/10.1111/j.1600-0870.2010.00456.x, 2010.

Hirdman, D., Sodemann, H., Eckhardt, S., Burkhart, J. F., Jefferson, A., Mefford, T., Quinn, P. K., Sharma, S., Ström, J., and Stohl, A.: Source identification of short-lived air pollutants in the Arctic using statistical analysis of measurement data and particle dispersion model output, Atmospheric Chemistry and Physics, 10, 669–693, https://doi.org/10.5194/acp-10-669-2010, 2010.

Johnsen, S. J., Dansgaard, W., and White, J. W. C.: The origin of Arctic precipitation under present and glacial conditions, Tellus B, 41B, 452–468, https://doi.org/https://doi.org/10.1111/j.1600-0889.1989.tb00321.x, 1989.

Keune, J., Schumacher, D. L., and Miralles, D. G.: A unified framework to estimate the origins of atmospheric moisture and heat using Lagrangian models, Geoscientific Model Development, 15, 1875–1898, https://doi.org/10.5194/gmd-15-1875-2022, 2022.



**Table C3.** Sectorisation variables contained in the series files.

| Variable | Unit |
| --- | --- |
| time | days since 0001-01-01 00:00:00 |
| sector | sector number |
| sector_area | km$^2$ |
| land_fraction_moisture_uptakes_boundary_layer | % |
| sea_fraction_moisture_uptakes_boundary_layer | % |
| land_fraction_moisture_uptakes_free_troposphere | % |
| sea_fraction_moisture_uptakes_free_troposphere | % |
| land_fraction_trajectory_location | % |
| sea_fraction_trajectory_location | % |
| land_fraction_moisture_transport | % |
| sea_fraction_moisture_transport | % |
| land_fraction_airmass_mixing | % |
| sea_fraction_airmass_mixing | % |
| land_fraction_rainout | % |
| sea_fraction_rainout | % |
| land_fraction_evaporation_minus_precipitation | % |
| sea_fraction_evaporation_minus_precipitation | % |

Laederach, A.: Characteristic scales of atmospheric moisture transport, Ph.D. thesis, ETH Zurich, Diss ETH No.23586,
https://doi.org/10.3929/ethz-a-010741025, 2016.

Laederach, A. and Sodemann, H.: A revised picture of the atmospheric residence time of water vapour, Geophys. Res. Letters, 43, 924–933,
https://doi.org/10.1002/2015GL067449, 2016.

Martius, O., Sodemann, H., Joos, H., Pfahl, S., Winschall, A., Croci-Maspoli, M., Graf, M., Madonna, E., Mueller, B., Schemm, S., Sedlacek,
J., Sprenger, M., and Wernli, H.: The role of upper-level dynamics and surface processes for the Pakistan flood in July 2010, Quart. J.
Royal Meteorol. Soc., https://doi.org/10.1002/qj.2082, 2012.

Masson-Delmotte, V., Buiron, D., Ekaykin, A., Frezzotti, M., Gallée, H., Jouzel, J., Krinner, G., Landais, A., Motoyama, A., Oerter, H., Pol,
K., Pollard, D., Ritz, C., Schlosser, E., Sime, L. C., Sodemann, H., Stenni, B., R., U., and Vimeux, F.: A comparison of the present and
last interglacial periods in six Antarctic ice cores, Clim. Past, 7, 397–423, 2011.

Miltenberger, A. K., Pfahl, S., and Wernli, H.: An online trajectory module (version 1.0) for the nonhydrostatic numerical weather prediction
model COSMO, Geoscientific Model Development, 6, 1989–2004, https://doi.org/10.5194/gmd-6-1989-2013, 2013.

Numaguti, A.: Origin and recycling processes of precipitating water over the Eurasian continent: Experiments using an atmospheric general
circulation model, J. Geophys. Res., 104, 1957–1972, http://dx.doi.org/10.1029/1998JD200026, 1999.

Osman, M. B., Smith, B. E., Trusel, L. D., Das, S. B., McConnell, J. R., Chellman, N., Arienzo, M., and Sodemann, H.: Abrupt Common Era
hydroclimate shifts drive west Greenland ice cap change, Nature Geoscience, 14, 756–761, https://doi.org/10.1038/s41561-021-00818-w,
900    2021.





Pfahl, S. and Sodemann, H.: What controls deuterium excess in global precipitation?, Climate of the Past, 10, 771–781, https://doi.org/10.5194/cp-10-771-2014, 2014.

Pisso, I., Sollum, E., Grythe, H., Kristiansen, N. I., Cassiani, M., Eckhardt, S., Arnold, D., Morton, D., Thompson, R. L., Groot Zwaaftink, C. D., Evangeliou, N., Sodemann, H., Haimberger, L., Henne, S., Brunner, D., Burkhart, J. F., Fouilloux, A., Brioude, J., Philipp, A.,
Seibert, P., and Stohl, A.: The Lagrangian particle dispersion model FLEXPART version 10.4, Geoscientific Model Development, 12, 4955–4997, https://doi.org/10.5194/gmd-12-4955-2019, 2019.

Salati, E., Dall'Olio, A., Matsui, E., and Gat, J. R.: Recycling of water in the Amazon Basin: An isotopic study, Water Resources Research, 15, 1250–1258, https://doi.org/https://doi.org/10.1029/WR015i005p01250, 1979.

Sodemann, H.: Beyond turnover time: Constraining the lifetime distribution of water vapor from simple and complex approaches, Journal of
the Atmospheric Sciences, 77, 413–433, https://doi.org/10.1175/JAS-D-18-0336.1, 2020.

Sodemann, H.: Particle position output from the FLEXPART dispersion model for a test case for use with the WaterSip moisture source diagnostic, [Dataset], https://doi.org/10.5281/zenodo.14836549, 2025.

Sodemann, H. and Stohl, A.: Asymmetries in the moisture origin of Antarctic precipitation, Geophys. Res. Lett., 36, L22 803, https://doi.org/10.1029/2009GL040242, 2009.

Sodemann, H. and Zubler, E.: Seasonal and inter-annual variability of the moisture sources for Alpine precipitation during 1995-2002, Int. J. Climatol., 30, 947–961, https://doi.org/10.1002/joc.1932, 2010.

Sodemann, H., Masson-Delmotte, V., Schwierz, C., Vinther, B. M., and Wernli, H.: Interannual variability of Greenland winter precipitation sources: 2. Effects of North Atlantic Oscillation variability on stable isotopes in precipitation, J. Geophys. Res., 113, D12 111, http://dx.doi.org/10.1029/2007JD009416, 2008a.

Sodemann, H., Schwierz, C., and Wernli, H.: Interannual variability of Greenland winter precipitation sources: Lagrangian moisture diagnostic and North Atlantic Oscillation influence, J. Geophys. Res., 113, D03 107, http://dx.doi.org/10.1029/2007JD008503, 2008b.

Sodemann, H., Schwierz, C., and Wernli, H.: Inter-annual variability of Greenland winter precipitation sources. Lagrangian moisture diagnostic and North Atlantic Oscillation influence, J. Geophys. Res., 113, D03 107, https://doi.org/10.1029/2007JD008503, 2008c.

Sprenger, M. and Wernli, H.: The LAGRANTO Lagrangian analysis tool – version 2.0, Geoscientific Model Development, 8, 2569–2586,
https://doi.org/10.5194/gmd-8-2569-2015, 2015.

Steen-Larsen, H. C., Sveinbjörnsdottir, A. E., Jonsson, T., Ritter, F., Bonne, J.-L., Masson-Delmotte, V., Sodemann, H., Blunier, T., Dahl-Jensen, D., and Vinther, B. M.: Moisture sources and synoptic to seasonal variability of North Atlantic water vapor isotopic composition, J. Geophys. Res., 120, 5757—-5774, https://doi.org/10.1002/ 2015JD023234, 2015.

Stohl, A.: Computation, accuracy and applications of trajectories–A review and bibliography, Atmos. Environ., 32, 947–966, http://www.
sciencedirect.com/science/article/pii/S1352231097004573, 1998.

Stohl, A.: Characteristics of atmospheric transport into the Arctic troposphere, Journal of Geophysical Research: Atmospheres, 111, https://doi.org/https://doi.org/10.1029/2005JD006888, 2006.

Stohl, A. and James, P.: A Lagrangian analysis of the atmospheric branch of the global water cycle. Part I: Method description, validation, and demonstration for the August 2002 flooding in Central Europe, J. Hydrometeorol., 5, 656–678, https://doi.org/10.1175/1525-
7541(2004)005<0656:ALAOTA>2.0.CO;2, 2004.

Stohl, A., Forster, C., Frank, A., Seibert, P., and Wotawa, G.: Technical note: The Lagrangian particle dispersion model FLEXPART version 6.2, Atmos. Chem. Phys., 5, 2461–2474, 2005.



Terpstra, A., Gorodetskaya, I. V., and Sodemann, H.: Linking Sub-Tropical Evaporation and Extreme Precipitation Over East Antarctica: An Atmospheric River Case Study, Journal of Geophysical Research: Atmospheres, 126, e2020JD033 617, https://doi.org/https://doi.org/10.1029/2020JD033617, e2020JD033617 2020JD033617, 2021.

Trenberth, K. E., Smith, L., Qian, T., Dai, A., and Fasullo, J.: Estimates of the Global Water Budget and Its Annual Cycle Using Observational and Model Data, Journal of Hydrometeorology, 8, 758 – 769, https://doi.org/10.1175/JHM600.1, 2007.

van der Ent, R. J., Tuinenburg, O. A., Knoche, H. R., Kunstmann, H., and Savenije, H. H. G.: Should we use a simple or complex model for moisture recycling and atmospheric moisture tracking?, Hydrol. Earth Syst. Sci., 17, 4869–4884, 2013.

van der Ent, R. J., Wang-Erlandsson, L., Keys, P. W., and Savenije, H. H. G.: Contrasting roles of interception and transpiration in the hydrological cycle - Part 2: Moisture recycling, Earth Sys. Dyn. Diss., 5, 281–326, https://doi.org/10.5194/esdd-5-281-2014, 2014.

Škerlak, B., Sprenger, M., and Wernli, H.: A global climatology of stratosphere–troposphere exchange using the ERA-Interim data set from 1979 to 2011, Atmospheric Chemistry and Physics, 14, 913–937, https://doi.org/10.5194/acp-14-913-2014, 2014.

Wang, Y., Sodemann, H., Hou, S., Masson-Delmotte, V., Jouzel, J., and Pang, H.: Snow accumulation and its moisture origin over Dome Argus, Antarctica, Climate Dynamics, 40, 731–742, https://doi.org/10.1007/s00382-012-1398-9, 2013.

Weng, Y., Johannessen, A., and Sodemann, H.: High-resolution stable isotope signature of a land-falling Atmospheric River in southern Norway, Weather and Climate Dynamics, 2, 713–737, https://doi.org/https://doi.org/10.5194/wcd-2-713-2021, 2021.

Winkler, R., Landais, A., Sodemann, H., Dümbgen, L., Priéa, F., Masson-Delmotte, V., Stenni, B., and Jouzel, J.: Deglaciation records of 17O-excess in East Antarctica: reliable reconstruction of oceanic relative humidity from coastal sites, Clim. Past, 8, 1–16, 2012.

Winschall, A., Pfahl, S., Sodemann, H., and Wernli, H.: Comparison of Eulerian and Lagrangian moisture source diagnostics – the flood event in eastern Europe in May 2010, Atmos. Chem. Phys., 14, 6605–6619, https://doi.org/10.5194/acp-14-6605-2014, 2014a.

Winschall, A., Sodemann, H., Pfahl, S., and Wernli, H.: How important is intensified evaporation for Mediterranean precipitation extremes?, J. Geophys. Res., 119, 5240—5256, https://doi.org/10.1002/2013JD021175, 2014b.

Yoshimura, K., Oki, T., Ohte, N., and Kanae, S.: Colored Moisture Analysis Estimates of Variations in 1998 Asian Monsoon Water Sources, Journal of the Meteorological Society of Japan, 82, 1315–1329, https://doi.org/10.2151/jmsj.2004.1315, 2004.

Zannoni, D., Steen-Larsen, H. C., Peters, A. J., Wahl, S., Sodemann, H., and Sveinbjörnsdóttir, A. E.: Non-Equilibrium Fractionation Factors for D/H and 18O/16O During Oceanic Evaporation in the North-West Atlantic Region, Journal of Geophysical Research: Atmospheres, 127, e2022JD037 076, https://doi.org/https://doi.org/10.1029/2022JD037076, e2022JD037076 2022JD037076, 2022.





**Figure B1.** Pre-defined sectorisations that are currently available in WaterSip. (a) NEEM sectorisation (11 sectors), (b) Antarctica sectorisation (21 sectors), (c) Borneo sectorisation (14 sectors), (d) Norway sectorisation (8 sectors), (e) Arctic sectorisation (21 sectors), (f) Belize sectorisation (15 sectors), (g) Alps sectorisation (9 sectors), (h) Pakistan sectorisation (6 sectors), (i) Latitude sectorisation (18 Sectors), (j) Longitude sectorisation (36 sectors), (k) China sectorisation (19 sectors), (l) Scandinavia sectorisation (23 sectors), (m) Arctic Seas sectorisation (19 sectors), (n) Pakistan 2022 sectorisation (13 sectors). Colour bar symbolic.