# Peer review of "The Lagrangian moisture source and transport diagnostic WaterSip V3.2"

_EGUsphere, 2025_

## Author Comment (AC2)

**Reply to review of "The Lagrangian moisture source and transport diagnostic WaterSip V3.2", submitted to GMD by H. Sodemann**

**Reviewer #1**

*This manuscript presents a comprehensive update and documentation of the WaterSip software (version 3.2), a diagnostic tool for identifying moisture sources and transport pathways associated with precipitation and atmospheric water vapour. The tool implements the Lagrangian diagnostic framework originally introduced by Sodemann et al. (2008), and offers support for trajectory data from models such as LAGRANTO and FLEXPART.*

*The manuscript is technically detailed and contains extensive explanations of the algorithmic structure, parameter configuration, example case setup, and diagnostic outputs. It represents a valuable contribution to the hydrometeorological community, particularly those using Lagrangian methods for moisture tracking. However, there are several areas where the manuscript could be improved significantly, especially in the following aspects:*

- *Validation of diagnostic results through comparison with observations or alternative algorithms;*

- *Technical clarity on certain assumptions and limitations;*

- *Demonstration of robustness and sensitivity through more systematic experiments;*

- *Improved structure and clarification of key terminology for broader accessibility.*

*I recommend major revisions before this manuscript is accepted for publication.*

**Reply:** Thank you for your detailed and constructive review. The specific comments are addressed below.

*1. Lack of Model Validation and Performance Benchmarking*

*While the algorithmic principles of WaterSip are well-founded, the manuscript lacks quantitative validation of the diagnostic results. In particular:*

- *No comparison with independent observational datasets (e.g., precipitation from GPM/IMERG or ERA5 reanalysis P);*

- *No benchmarking against other Lagrangian diagnostics, such as WAM-2layers, FLEXPART-WATER, or isotope-enabled models (e.g., COSMOiso);*

- *The Lagrangian precipitation estimate $P\sim\tilde{P}P\sim$ is claimed to have an error of 20–30%, yet this is not demonstrated empirically in the paper.*

*Recommendation: Include a comparison of WaterSip-derived precipitation estimates and source regions against satellite/reanalysis precipitation and/or results from other established methods. This would help quantify accuracy and justify the use of default parameters (e.g., RHc, $\Delta q$ thresholds).*

**Reply:** The reviewer suggests to add more material that compares the Lagrangian precipitation estimate with other datasets, as well as a comparison of results from the diagnostic to other methods. There are a number of reasons why I would prefer to strongly limit this kind of analysis in the present manuscript:

- Comparison to independent observational datasets: WaterSip's primary task is not to provide precipitation, but to detect precipitation properties. While a comparison to the precipitation in the reanalysis dataset itself is useful to learn about the representativeness of the diagnostic results, a comparison to other observational or blended datasets is beyond the scope of this study. For that purpose, it would be more useful to directly use studies that compare between reanalysis data and observational/merged precipitation itself. This will be further clarified in the revised manuscript, including the addition of relevant references to comparisons between different precipitation datasets, and possibly the addition of a reanalysis precipitation map for the case study.

- Benchmarking against Lagrangian diagnostics and isotope enabled models: The manuscript shows a comparison to the results from the Stohl and James (2004) method (Fig. 2d). Including a comparison to other models either for one case study, or several case studies, is in my view far beyond of what can fit into

the scope of this manuscript - if it should be done in a way that it becomes sufficiently meaningful. Without objective 'truth' about the moisture sources at hand, it only possible to make relative comparisons. There are several papers published in the literature that attempt such comparisons, and other efforts are currently ongoing. Where possible, additional citations will be added in the revised manuscript, and the challenges of finding moisture source properties from different methods will be mentioned when discussing the case study.

• The Lagrangian precipitation estimate has been introduced by James et al. (2004) and Stohl and James (2004). The 20-30 % is a ballpark number, but results will vary depending on study region. Several studies have shown that there can be over- or underestimation, and some studies that are cited have systematically evaluated dependency on thresholds. Users will have to make their own evaluations depending on thresholds and regions that are studied. It is an important aspect throughout the manuscript that users need to be aware of what they are doing. This perspective will be underlined in the revised manuscript, and potentially backed up by a comparison with reanalysis precipitation.

**2. Insufficient Sensitivity Experiments**

*The diagnostic depends heavily on multiple user-defined thresholds, such as:*

• *Moisture uptake threshold ($\Delta qc$),*

• *Precipitation threshold ($\Delta qp$),*

• *Critical relative humidity (RHc),*

• *Trajectory length (L) and time step ($\Delta t$),*

• *Boundary-layer height scale (sh).*

*While some default values are provided, the manuscript does not present any systematic sensitivity tests to justify these defaults or examine result variability.*

*Recommendation: Provide at least one sensitivity experiment (e.g., with RHc = 60%, 80%, 90% or $\Delta qc$ = 0.1, 0.2, 0.3 g/kg/6h) using the Scandinavia case to demonstrate how output fields (e.g., source footprints, $\tilde{P}$) are affected. This will help users understand uncertainty and robustness.*

**Reply:** It is important to note that there is a large body of literature where this method was applied from a time before code was commonly published open access, and where these thresholds have been thoroughly tested. Instead of redoing such previous work in this manuscript for a limited case study, my primary choice has been to use literature references. However, I agree that a limited sensitivity experiment will be useful to include for the Scandinavia case, support users in running their own sensitivity experiments.

**3. Ambiguity in Treatment of Mixing vs. Precipitation**

*The distinction between moisture losses due to precipitation vs. dry mixing is briefly described but remains ambiguous in practical terms:*

• *How are "mixing events" defined and treated in the accounting algorithm?*

• *Are they excluded from precipitation source attribution entirely?*

• *How does this impact attribution over dry regions or under sub-saturated conditions?*

*Recommendation: Include a dedicated subsection clarifying how dry mixing events are separated and whether/how they influence the fractional contribution calculation. Provide a sample output or visualization that isolates these cases.*

**Reply:** Identified mixing events are shown in Fig. 4c, the current manuscript text incorrectly refers to Fig. 2c. In the revised manuscript, the text describing mixing events will be extended and clarified.

**4. Limited Scope of Case Study**

*The case study over Scandinavia is informative but lacks depth and generality:*

- *It only covers a short period (10–20 Aug 2022) with one configuration;*
- *There is no validation of the Lagrangian precipitation estimates against ERA5 or in-situ observations;*
- *The transport features are discussed qualitatively without statistical summaries (e.g., source region contributions by %).*

*Recommendation:*

- *Add a second case study (e.g., a winter event or tropical cyclone) to demonstrate versatility;*
- *Include plots/tables showing the percent contribution of major source regions (e.g., local vs. oceanic);*
- *Overlay gridded WaterSip P~\tilde{P}P~ with observational data (e.g., E-OBS or GPCC).*

**Reply:** The function of the Scandinavia case study is to help users of WaterSip getting started with a concrete example. A second case study would not add to that purpose, but potentially confuse and tire readers due to the increased manuscript length. There is a large body of literature available with different geographical foci and for different time periods (Table 1), and in the revision this role of the case study will be further clarified.

A comparison between the Lagrangian precipitation estimate and ERA5 precipitation will be useful to add to illustrate the differences between both quantities. It will also be clarified again in the text that the purpose of the WaterSip diagnostic is not to estimate precipitation, but to identify precipitation properties. I consider therefore a comparison to observational datasets will be beyond the scope of this GMD paper.

Finally, a brief summary table summarising the major source regions will be added in the revised manuscript.

**5. No Performance or Computational Cost Analysis**

*Given the tool is designed for high-volume Lagrangian data, its computational performance, memory usage, and scalability are essential for practical adoption:*

*Recommendation: Add a short section or table reporting:*

- *Typical runtime and memory usage for the example case;*
- *Speedup with OpenMP threads;*
- *Bottlenecks or limitations for large-scale usage.*

**Reply:** A brief paragraph describing computational performance and memory usage for the test case will be added in the revised manuscript.

**Minor Comments & Suggestions**

1. *Clarify terminology early (Section 1):*
   - *Define "uptake", "accounting", "residual moisture", and "arrival grid" explicitly.*
   - *Consider a graphical workflow diagram.*
2. *Equations (6)–(9):*
   - *Include variable definitions in-line with the equations, especially for readers not familiar with the 2008 method.*
3. *Section 2.5: Too long and fragmented. Suggest splitting into:*
   - *"Core algorithm parameters" ($\Delta q$, RHc, sh),*
   - *"Grid and output configuration",*
   - *"Optional diagnostics".*
4. *Figures:*
   - *Add scale bars and legends (e.g., units in mm/day);*

- *Some figures lack clarity (e.g., Fig. 2d – difficult to read e-p shading);*
- *Add observational overlay for better interpretation.*

5. *Code availability: Ensure a DOI or stable link is provided. Consider creating a GitHub/Zenodo archive.*

6. *Language & Style:*
   - *Mostly clear, but some long and nested sentences in Section 2–3 could be simplified.*
   - *Example: "Air parcels will only be retained for analysis…" → split into clearer bullet rules.*

**Reply:** These minor and technical comments will be implemented as suggested.

**Conclusion**

*The manuscript presents a valuable and much-needed technical documentation of WaterSip V3.2 and the Lagrangian moisture source diagnostic algorithm. However, to be suitable for publication in a journal such as GMD or HESS Discussions, the following critical issues must be addressed:*

- *Quantitative validation of results,*
- *Sensitivity and uncertainty analysis,*
- *Clear treatment of physical assumptions (e.g., mixing vs. precipitation),*
- *Extended and comparative case studies.*

**Reply:** all comments have been addressed in the above.

**References:**

James, P., Stohl, A., Spichtinger, N., Eckhardt, S., and Forster, C.: Climatological aspects of the extreme European rainfall of August 2002 and a trajectory method for estimating the associated evaporative source regions, Nat. Hazards Earth Syst. Sci., 4, 733–746, https://doi.org/10.5194/nhess-4-733-2004, 2004.

Stohl, A., and P. James, 2004: A Lagrangian Analysis of the Atmospheric Branch of the Global Water Cycle. Part I: Method Description, Validation, and Demonstration for the August 2002 Flooding in Central Europe. J. Hydrometeor., 5, 656–678, https://doi.org/10.1175/1525-7541(2004)005<0656:ALAOTA>2.0.CO;2.

---

## Author Comment (AC3)

**Reply to review of "The Lagrangian moisture source and transport diagnostic WaterSip V3.2", submitted to GMD by H. Sodemann**

**Reviewer #2**

*This paper describes the software WaterSip, a widely used Lagrangian moisture source diagnostic that is based on the accounting algorithm developed by Sodemann et al. (2008). Apart from diagnosing moisture sources for precipitation or vapor, WaterSip can provide information on moisture source conditions, transport, and arrival quantities. The paper gives an overview over the diagnostic method, describes how to configure and run WaterSip, and explains the different output files and variables using a test case in Scandinavia as an example. It also briefly discusses potential errors and uncertainties in the results.*

*WaterSip is a very powerful tool, but so far has been difficult to use due to restricted access and limited documentation. This paper will greatly enhance the accessibility and usability of WaterSip. It is well-written and nicely structured, providing useful guidelines for potential users. I also very much like the idea of providing a test case with all the necessary input and configuration files for running WaterSip, which is a good starting point for new WaterSip users. There are some problems related to the test case, but they can hopefully be fixed. Apart from that my comments are mostly minor, and I recommend publication after addressing those.*

**Reply:** Thank you for the thorough and constructive review. All comments are addressed below.

***General comments***

*1) I tried to run WaterSip with the provided test case, but it did not really work. During compilation it first did not find the netcdfcpp.h file, and I think this is because there is a mistake in the makefile. The NETCDFINC path should be added to the compilation step instead of the linking step:*

```
COMPILE = $(CC) $(CFLAGS) -c $(NETCDFINC)
FLINK = $(NETCDFLIB) -lnetcdf_c++4 -lnetcdf_c++ -lnetcdf -lsz -lz -ldl -lm
-lpthread -lcurl -lstdc++ -fopenmp
```

*During runtime there were some other problems:*
- *The startDate in the input file is after the last shortposit file provided on Zenodo. It should be changed to 20220811-000000 (or alternatively more shortposit files should be provided).*
- *The particle number maxPart is too low, I got the error "***ERROR: could not assign particle, maximum particle number exceeded!". With maxPart = 100000001 it worked.*
- *The reading of the input file ended up in an infinite loop in Parser::skipBlanks. I had to add filestr.get() on line 688 in Parser.cpp for it to work.*
- *After completing 92.2%, WaterSip crashed with a segmentation fault. I did not figure out why.*

**Reply:** I'm sorry about the trouble getting the software to compile and run, and thank you for testing out some solutions. Indeed, the makefile is currently not sufficiently system independent. In fact, the solution is to add the library path to the netCDF-C++ library include files to the makefile, then the netcdfcpp.h file can be found. An updated makefile will be made available with the resubmission.

Regarding the runtime problems, it appears that the wrong INPUT file was included in the archive, where the settings don't match the test data. In addition, the Parser.cpp routine was not sufficiently robust to handle additional blank characters that had been added to the input file for nicer formatting. An updated version of the Parser.cpp file will be made available with the source code.

Furthermore, to facilitate running the test case, a bash script will be made available with WaterSip that will download the test case, build the file structure implied in the INPUT file, and thus allow users to more easily run the test case.

*2) A new version of FLEXPART has been released recently (FLEXPART11, Bakels et al., 2024). Does WaterSip work with this new version as well, or only the 10.4 version? Since the new version writes trajectory output to NetCDF files, it would probably require rewriting the routines reading the input data. However, it might be worth it because FLEXPART11 has several advantages compared to FLEXPART10.4, for example (relevant for Waterip) improved trajectory accuracy and the option to write out average instead of instantaneous values along trajectories.*

**Reply:** This is an interesting question. Currently, WaterSip is not able to read NetCDF files from FLEXPART11. The simplest way to make this work is to create a post-processor that will convert netCDF FLEXPART11 particle files to the binary dump format used with FLEXPART10.4 and before. This information will be added to the revised manuscript.

*3) The section on errors and uncertainties is very short, but I think it is an important section and should be extended a bit. I would suggest to add a few figures showing the sensitivity of the results for the Scandinavia case to the settings, specifically the timeStep, the uptakeThreshold and precipThreshold, and arrivalRHMin. This would be very useful for new users to understand the influence of these settings. Also some potential error sources are currently not mentioned in this section, e.g. the fact that WaterSip always assumes either only moisture uptakes or only moisture losses during one timestep but not both. This could lead to an overestimation of remote and an underestimation of local moisture sources. Or, when using WaterSip as a diagnostic for surface evaporation, the assumption that water evaporated from the location (lat,lon) where the moisture is taken up by the air parcel might not always be true.*

**Reply:** A similar comment has been made by the other reviewer. While my general approach has been to refer to systematic sensitivity tests in earlier studies, I agree that it will be useful to include some sensitivity results for the test case in this manuscript. The additional error sources mentioned by the reviewer will also added in the revised manuscript.

*4) I did not fully understand the (difference between the) moisture source and transport quantities. For the moisture source quantities, the values are multiplied by $\Delta q * f * m$, for the transport quantities, they are multiplied only by $f * m$. I think the problem is that I don't really know what $f$ and $\Delta q$ are in this case. Are these the values after the accounting? If so, isn't $f$ exactly $\Delta q/q\_0$? So why is then $f$ multiplied by $\Delta q$ again? I am sure this is all done correctly in the code, but it could be explained a bit better (see also my specific comments on Equations 19 & 20).*

**Reply:** The difference between source and transport quantities is that the weighting of the source quantities is only done once for the final evaluated contribution from each source region, whereas the transport quantities are also gridded at each time step during the airmass transport. Regarding the question why $\Delta q * f * m$ is calculated rather than $f * m$ only, in the first case, the resulting quantity is the accounted mass of water (since m is a constant, giving the air mass of the particle).

**Specific comments**

*L9-11: This is a repetition of L2-4*

*L22: Bracket around Stohl and James (2004)*

*L74: chose -> choose*

*L108: Figure reference broken*

*L133: doe -> does*

*L151: What do you mean by „in case trajectories are used rather than air parcels"?*

*L154: I think this first sentence is not needed here.*

**Reply:** These items will be corrected in the revised manuscript

*Equation 12: Shouldn't this be the sum over i?*

*L168: This is interesting. Why does this (f_tot > 1) happen? This would be a case where the algorithm from Dütsch et al. (2018) is not equivalent, because there f_tot is by definition always <= 1.*

**Reply:** Yes, the summation should be over i. The f_tot > 1 can happen due to the delta_q_c threshold. This will be clarified in the revised manuscript.

*L233: Why „differences in"? Not just the source contributions themselves?*

*L266: Do you mean atmospheric properties? Because the positions come from the trajectory calculation, so that would be the second part of the sentence.*

**Reply:** These comments will be addressed by rephrasing the respective sentences.

*L314: What would be a reason for setting a maximum relative humidity?*

**Reply:** A hypothetical example would be in the tracing of vapour rather than precipitation properties, where one could be interested in cloud-free arrival locations only.

*L323: partcels -> parcels*

*L360-369: I didn't understand this part with the gridRadius.*

*L395: Could you briefly explain what these methods do, specifically Gustafsson et al. (2010) and Dirmeyer and Brubaker (1999)?*

*L403: Remove „are"*

*L436: Closing bracket missing*

*L467: chose -> choose*

*L483: Maybe start with a brief description of the meteorological situation for the event?*

*L498 (and others): Sometimes day is d, dy, or day. Please be consistent.*

*L515: Stohl et al. (2005) or Stohl and James (2004)?*

**Reply:** These comments will be addressed by rephrasing the respective sentences.

*L539-L545: I don't understand how the quantities moisture transport, air mass mixing, and rainout are obtained. Could you explain this better? For example, what is meant by gridded product of the specific humidity?*

**Reply:** The explanation of transport quantities will be rephrased and extended in the revised manuscript.

*L551: I don't understand this first sentence. Do you mean differences _in_ moisture sources and transport? But moisture sources are not shown here… (?)*

*L554: quantity -> quantities*

**Reply:** These comments will be addressed by rephrasing the respective sentences.

*L554-L557: How does the precipitation estimate by WaterSip compare to ERA5 precipitation? This would be a good validity check.*

**Reply:** See reply to major comment #3

*Equation 19: Shouldn't the denominator be P_tot (without k)? And the enumerator would correspond to T^k_0 * P^k_tot?*

**Reply:** Correct, thank you for pointing this out. This will be corrected and clarified in the revision.

*L579: Earlier λ and φ were used for lon and lat. Please use consistent notation throughout the manuscript. How is the mean over longitudes calculated? Are the coordinates converted to a Cartesian grid first? Otherwise there would be problems for e.g. lon1=-179 and lon2=179.*

*Equation 20: What is M? I assume the time steps of the trajectories? You could use this also in Equation 12 for consistency. Shouldn't Δq also appear in the denominator?*

**Reply:** Yes, the longitude averaging is done using conversion to a Cartesian grid. The notation will be adjusted to be consistent throughout as suggested.

*L583: zeta -> \zeta*

*L595: This is a detail, but wouldn't it be better to use the land fraction itself in the averaging (instead of 0 or 1)?*

*L600: and Eq. 19*

*L604 and L617: „the" too much.*

*L607: Maybe briefly explain what d-excess is or define it?*

*Equation 22: Is N the same as M in Equation 20? If so, use M instead?*

*L650: where -> were*

*L698: Would it be possible to provide a script for reading the .traj files? Otherwise this is difficult because of the binary format.*

*L763: arrival _in the_ target region?*

*L780: The link to the WaterSip source code does not work. The one given on Zenodo is correct.*

**Reply:** These comments will be addressed by editing the respective sentences and equations.

**Figure and table comments**

*Figures 5,6,7: I would use a different colormap. This one looks like topography and is not very intuitive for sequential values (e.g. white means high).*

**Reply:** I will try to find an alternative, suitable colormap for the revised manuscript.

*Figure 9: The locations of the colors don't correspond to the labels of the colorbar. Which one is correct?*

**Reply:** The labels have been flipped in the figure. This will be corrected, as well as potentially using a projection that shows locations north of 80ºN.

*Figure 10: Are these quantities weighted by f? From Figure 10 it seems so, but the text doesn't mention it. In (d) where can we see the time before arrival mentioned in the caption?*

**Reply:** Yes, the text will be updated accordingly. The caption to (d) gives the PDF as a function of distance, which will be corrected.

*Table 3: Why do the uptakes have negative values?*

**Reply:** The negative quantity is actually time before arrival in hours, rather than uptake, which is the gridded quantity. The table will be adjusted to make this clear in the revised manuscript.

*Figure 11b: I am not sure what is shown here. The figure says specific humidity, but the text says total accounted fraction. Where do we see that „moisture uptakes of more than 10 days before arrival are mostly overwritten by later uptake events"? I would suggest to use a different colormap, because red for moist and blue for wet is a bit counterintuitive. Also jet has many other problems (irregular lightness, not perceptually uniform, not colorblind-friendly).*

**Reply:** The figure gives the correct information, the text will be corrected in the revised manuscript. I will consider using a different colormap.

**References**

*Bakels, L., Tatsii, D., Tipka, A., Thompson, R., Dütsch, M., Blaschek, M., Seibert, P., Baier, K., Bucci, S., Cassiani, M., Eckhardt, S., Groot Zwaaftink, C., Henne, S., Kaufmann, P., Lechner, V., Maurer, C., Mulder, M. D., Pisso, I., Plach, A., Subramanian, R., Vojta, M., and Stohl, A.: FLEXPART version 11: improved accuracy, efficiency, and flexibility, Geoscientific Model Development, 17, 7595–7627, https://doi.org/10.5194/gmd-17-7595-2024, 2024.*

*Dütsch, M., Pfahl, S., Meyer, M., and Wernli, H.: Lagrangian process attribution of isotopic variations in near-surface water vapour in a 30-year regional climate simulation over Europe, Atmospheric Chemistry and Physics, 18, 1653–1669, https://doi.org/10.5194/acp-18-1653-2018, 2018.*

*Sodemann, H., Schwierz, C., and Wernli, H.: Interannual variability of Greenland winter precipitation sources: Lagrangian moisture diagnostic and North Atlantic Oscillation influence, J. Geophys. Res., 113, D03 107, http://dx.doi.org/10.1029/2007JD008503, 2008.*

---

## Author Response (AR1)

**Reply to reviews of "The Lagrangian moisture source and transport diagnostic WaterSip V3.2", submitted to GMD by H. Sodemann**

Bergen, 11.08.2025

Dear Dr. Maher,

I have now revised my submitted manuscript, taking all comments from reviewers into account, as detailed below. The comments from the two reviewers were very helpful in clarifying the overall message of the the manuscript and allowed to remove some errors and inconsistencies. Further analyses requested by the reviewers have lead to some restructuring and to an extention of the discussion section.

In addition, I have shifted the time period of the case study by 5 days to obtain a more clear delineation of the start and end of the event. The overall interpretation of the illustrative example case is still very much aligned with the original submission. I have also changed the units of two output quantities "particle mass density" and "moisture transport" in the computer code and updated Figure 4 and the writing accordingly. I think this change will make the overall description of how these quantities are obtained easier to follow, and more useful for interpretation.

The model code version has therefore received a minor increase from version 3.2.0 to 3.2.1. This version number detail does not change the title which only states the first two version indicators.

With best regards,

Harald Sodemann

**Reviewer #1**

*This manuscript presents a comprehensive update and documentation of the WaterSip software (version 3.2), a diagnostic tool for identifying moisture sources and transport pathways associated with precipitation and atmospheric water vapour. The tool implements the Lagrangian diagnostic framework originally introduced by Sodemann et al. (2008), and offers support for trajectory data from models such as LAGRANTO and FLEXPART.*

*The manuscript is technically detailed and contains extensive explanations of the algorithmic structure, parameter configuration, example case setup, and diagnostic outputs. It represents a valuable contribution to the hydrometeorological community, particularly those using Lagrangian methods for moisture tracking. However, there are several areas where the manuscript could be improved significantly, especially in the following aspects:*

· *Validation of diagnostic results through comparison with observations or alternative algorithms;*

· *Technical clarity on certain assumptions and limitations;*

· *Demonstration of robustness and sensitivity through more systematic experiments;*

· *Improved structure and clarification of key terminology for broader accessibility.*

*I recommend major revisions before this manuscript is accepted for publication.*

**Reply:** Thank you for your detailed and constructive review. The specific comments are addressed below.

*1. Lack of Model Validation and Performance Benchmarking*

*While the algorithmic principles of WaterSip are well-founded, the manuscript lacks quantitative validation of the diagnostic results. In particular:*

- *No comparison with independent observational datasets (e.g., precipitation from GPM/IMERG or ERA5 reanalysis P);*

- *No benchmarking against other Lagrangian diagnostics, such as WAM-2layers, FLEXPART-WATER, or isotope-enabled models (e.g., COSMOiso);*

- *The Lagrangian precipitation estimate P~\tilde{P}P~ is claimed to have an error of 20–30%, yet this is not demonstrated empirically in the paper.*

*Recommendation: Include a comparison of WaterSip-derived precipitation estimates and source regions against satellite/reanalysis precipitation and/or results from other established methods. This would help quantify accuracy and justify the use of default parameters (e.g., RHc, Δq thresholds).*

**Reply:** The reviewer suggests to add more material that compares the Lagrangian precipitation estimate with other datasets, as well as a comparison of results from the diagnostic to other methods. There are a number of reasons why I would prefer to strongly limit this kind of analysis in the present manuscript:

- Comparison to independent observational datasets: WaterSip's primary task is not to provide precipitation, but to detect precipitation properties. While a comparison to the precipitation in the reanalysis dataset itself is useful to learn about the representativeness of the diagnostic results, a comparison to other observational or blended datasets is beyond the scope of this study. For that purpose, it would be more useful to directly use studies that compare between reanalysis data and observational/merged precipitation itself. This will be further clarified in the revised manuscript, including the addition of relevant references to comparisons between different precipitation datasets, and possibly the addition of a reanalysis precipitation map for the case study.

- Benchmarking against Lagrangian diagnostics and isotope enabled models: The manuscript shows a comparison to the results from the Stohl and James (2004) method (Fig. 2d). Including a comparison to other models either for one case study, or several case studies, is in my view far beyond of what can fit into the scope of this manuscript - if it should be done in a way that it becomes sufficiently meaningful. Without objective 'truth' about the moisture sources at hand, it only possible to make relative comparisons. There are several papers published in the literature that attempt such comparisons, and other efforts are currently ongoing. These are already cited in relevant locations.

- The Lagrangian precipitation estimate has been introduced by James et al. (2004) and Stohl and James (2004). The 20-30 % is a ballpark number, but results will vary depending on study region. Several studies have shown that there can be over- or underestimation, and some studies that are cited have systematically evaluated dependency on thresholds. In the revised manuscript, it is now stated that the underestimation of precipitation is 17% for the chosen case as a domain average. In addition, the ERA5 precipitation is now overlayed in Fig. 5, added as time series in Fig. 8, and the precipitation patters within the domain are compared direcly in Figure A1.

- Users will have to make their own evaluations depending on thresholds and regions that are studied. It is an important aspect throughout the manuscript that users need to be aware of what they are doing. This perspective has been further underlined in the revised manuscript, and is also backed up by the comparison to reanalysis precipitation.

**2. Insufficient Sensitivity Experiments**

*The diagnostic depends heavily on multiple user-defined thresholds, such as:*

- *Moisture uptake threshold (Δqc),*

- *Precipitation threshold (Δqp),*

- *Critical relative humidity (RHc),*

- *Trajectory length (L) and time step (Δt),*

- *Boundary-layer height scale (sh).*

*While some default values are provided, the manuscript does not present any systematic sensitivity tests to justify these defaults or examine result variability.*

*Recommendation: Provide at least one sensitivity experiment (e.g., with RHc = 60%, 80%, 90% or $\Delta qc$ = 0.1, 0.2, 0.3 g/kg/6h) using the Scandinavia case to demonstrate how output fields (e.g., source footprints, $\tilde{P}$) are affected. This will help users understand uncertainty and robustness.*

**Reply:** It is important to note that there is a large body of literature where this method was applied from a time before code was commonly published open access, and where these thresholds have been thoroughly tested. Instead of redoing such previous work in this manuscript for a limited case study, my primary choice has been to use literature references. However, I agree that a limited sensitivity experiment can be useful for new users of the WaterSip software in this manuscript, supporting own sensitivity experiments. A sensitivity analysis for the Scandinavia case has been added regarding the paraeters $RH_c$, $\Delta q_c$, and trajectory length, as also suggested by reviewer #2. The results of the sensitivity analysis have been quantified in terms of the precipitation estimate, the total accounted fraction and the moisture source distance. The sensitivity experiments are discussed in the new Sec. 6.4 and summarized in Table 5.

**3. Ambiguity in Treatment of Mixing vs. Precipitation**

*The distinction between moisture losses due to precipitation vs. dry mixing is briefly described but remains ambiguous in practical terms:*

- *How are "mixing events" defined and treated in the accounting algorithm?*

- *Are they excluded from precipitation source attribution entirely?*

- *How does this impact attribution over dry regions or under sub-saturated conditions?*

*Recommendation: Include a dedicated subsection clarifying how dry mixing events are separated and whether/how they influence the fractional contribution calculation. Provide a sample output or visualization that isolates these cases.*

**Reply:** A similar comment had been made by reviewer #2. Identified mixing events are already shown in Fig. 4c, the current manuscript text incorrectly referred to Fig. 2c. In the revised manuscript, the text describing mixing events has been extended and clarified. In the current version of WaterSip, these mixing events have the same consequence for the source attribution as precipitation events and are simply singled out and written out separately from precipitation events. Future work could test if mixing events should be treated differently during the accounting.

**4. Limited Scope of Case Study**

*The case study over Scandinavia is informative but lacks depth and generality:*

- *It only covers a short period (10–20 Aug 2022) with one configuration;*

- *There is no validation of the Lagrangian precipitation estimates against ERA5 or in-situ observations;*

- *The transport features are discussed qualitatively without statistical summaries (e.g., source region contributions by %).*

*Recommendation:*

- *Add a second case study (e.g., a winter event or tropical cyclone) to demonstrate versatility;*

- *Include plots/tables showing the percent contribution of major source regions (e.g., local vs. oceanic);*

- *Overlay gridded WaterSip P~\tilde{P}P~ with observational data (e.g., E-OBS or GPCC).*

**Reply:** The function of the Scandinavia case study is to help users of WaterSip getting started with a concrete example. A second case study would not add to that purpose, but potentially confuse and tire readers due to the increased manuscript length. There is a large body of literature available with different geographical foci and for different time periods (Table 1), and in the revision this role of the case study will be further clarified.

A comparison between the Lagrangian precipitation estimate and ERA5 precipitation has been added in the revised manuscript to illustrate the differences between both quantities. It has also been emphasized again in the revised text that the purpose of the WaterSip diagnostic is not to estimate precipitation, but to identify precipitation properties. I consider a further comparison to observational datasets will be beyond the scope of this GMD paper.

The source region contributions for the case study were already shown in Fig. 9, but are now also shown in a new Table 3 and discussed briefly in the text. Note that the time period of the case study has been changed to 5-15 August 2022, since the previous period started in the middle of a mid-latitude cyclone being present in the study domain. All figures have been updated and the text has been revised accordingly.

**5. No Performance or Computational Cost Analysis**

*Given the tool is designed for high-volume Lagrangian data, its computational performance, memory usage, and scalability are essential for practical adoption:*

*Recommendation: Add a short section or table reporting:*

- *Typical runtime and memory usage for the example case;*
- *Speedup with OpenMP threads;*
- *Bottlenecks or limitations for large-scale usage.*

**Reply:** A brief paragraph has been added in the revised manuscript as new Section 5.3, describing the computational performance and memory usage of the diagnostic for the Scandinavia test case.

**Minor Comments & Suggestions**

1. *Clarify terminology early (Section 1):*
   - *Define "uptake", "accounting", "residual moisture", and "arrival grid" explicitly.*

**Reply:** The definition of key terminology such as "uptakes" and "moisture sources" has been revised and is now hopefully clearer.

   - *Consider a graphical workflow diagram.*

**Reply:** A simple graphical workflow diagram has been added to Figure 1 as panel (c). This panel is now discussed in Section 2.

2. *Equations (6)–(9):*
   - *Include variable definitions in-line with the equations, especially for readers not familiar with the 2008 method.*

**Reply:** revised as suggested.

3. *Section 2.5: Too long and fragmented. Suggest splitting into:*
   - *"Core algorithm parameters" ($\Delta q$, RHc, sh),*
   - *"Grid and output configuration",*
   - *"Optional diagnostics".*

**Reply:** These sections have been rearranged and split into sections 3 and 4, with new subtitles following the suggestions from the reviewer.

4. *Figures:*
   - *Add scale bars and legends (e.g., units in mm/day);*
   - *Some figures lack clarity (e.g., Fig. 2d – difficult to read e-p shading);*
   - *Add observational overlay for better interpretation.*

**Reply:** The figures have been improved in terms of consistency of the units, the background shading to facilitate reading Fig. 2d, and an ERA5 data overlay has been added to Fig. 5a.

5. *Code availability: Ensure a DOI or stable link is provided. Consider creating a GitHub/Zenodo archive.*

**Reply:** The updated model code and new test case dataset are available on two separate data repositories, each with their valid doi.

6. *Language & Style:*
   - *Mostly clear, but some long and nested sentences in Section 2–3 could be simplified.*
   - *Example: "Air parcels will only be retained for analysis…" → split into clearer bullet rules.*

**Reply:** This sentence has been rephrased.

**Conclusion**

*The manuscript presents a valuable and much-needed technical documentation of WaterSip V3.2 and the Lagrangian moisture source diagnostic algorithm. However, to be suitable for publication in a journal such as GMD or HESS Discussions, the following critical issues must be addressed:*

- *Quantitative validation of results,*
- *Sensitivity and uncertainty analysis,*
- *Clear treatment of physical assumptions (e.g., mixing vs. precipitation),*
- *Extended and comparative case studies.*

**Reply:** all comments have been addressed in the above.

**References:**

James, P., Stohl, A., Spichtinger, N., Eckhardt, S., and Forster, C.: Climatological aspects of the extreme European rainfall of August 2002 and a trajectory method for estimating the associated evaporative source regions, Nat. Hazards Earth Syst. Sci., 4, 733–746, https://doi.org/10.5194/nhess-4-733-2004, 2004.

Stohl, A., and P. James, 2004: A Lagrangian Analysis of the Atmospheric Branch of the Global Water Cycle. Part I: Method Description, Validation, and Demonstration for the August 2002 Flooding in Central Europe. J. Hydrometeor., 5, 656–678, https://doi.org/10.1175/1525-7541(2004)005<0656:ALAOTA>2.0.CO;2.

**Reviewer #2**

*This paper describes the software WaterSip, a widely used Lagrangian moisture source diagnostic that is based on the accounting algorithm developed by Sodemann et al. (2008). Apart from diagnosing moisture sources for precipitation or vapor, WaterSip can provide information on moisture source conditions, transport, and arrival quantities. The paper gives an overview over the diagnostic method, describes how to configure and run WaterSip, and explains the different output files and variables using a test case in Scandinavia as an example. It also briefly discusses potential errors and uncertainties in the results.*

*WaterSip is a very powerful tool, but so far has been difficult to use due to restricted access and limited documentation. This paper will greatly enhance the accessibility and usability of WaterSip. It is well-written and nicely structured, providing useful guidelines for potential users. I also very much like the idea of providing a test case with all the necessary input and configuration files for running WaterSip, which is a good starting point for new WaterSip users. There are some problems related to the test case, but they can hopefully be fixed. Apart from that my comments are mostly minor, and I recommend publication after addressing those.*

**Reply:** Thank you for the thorough and constructive review. All comments are addressed below.

***General comments***

*1) I tried to run WaterSip with the provided test case, but it did not really work. During compilation it first did not find the netcdfcpp.h file, and I think this is because there is a mistake in the makefile. The NETCDFINC path should be added to the compilation step instead of the linking step:*

```
COMPILE = $(CC) $(CFLAGS) -c $(NETCDFINC)
FLINK = $(NETCDFLIB) -lnetcdf_c++4 -lnetcdf_c++ -lnetcdf -lsz -lz -ldl -lm
-lpthread -lcurl -lstdc++ -fopenmp
```

*During runtime there were some other problems:*
- *The startDate in the input file is after the last shortposit file provided on Zenodo. It should be changed to 20220811-000000 (or alternatively more shortposit files should be provided).*
- *The particle number maxPart is too low, I got the error "\*\*\*ERROR: could not assign particle, maximum particle number exceeded!". With maxPart = 100000001 it worked.*
- *The reading of the input file ended up in an infinite loop in Parser::skipBlanks. I had to add filestr.get() on line 688 in Parser.cpp for it to work.*
- *After completing 92.2%, WaterSip crashed with a segmentation fault. I did not figure out why.*

**Reply:** I'm sorry about the trouble getting the software to compile and run, and thank you for testing out some solutions. Indeed, the makefile is currently not sufficiently system independent. In fact, the solution is to add the library path to the netCDF-C++ library include files to the makefile, then the netcdfcpp.h file can be found. An updated makefile has beenmade available in the source code.

Regarding the runtime problems, it appears that the wrong INPUT file was included in the archive, where the settings don't match the test data. In addition, the Parser.cpp routine was not sufficiently robust to handle

additional blank characters that had been added to the input file for nicer formatting. An updated version of the Parser.cpp file has been made available with the source code. The version number of WaterSip has been changed to 3.2.1. The software crash seems to be related to the input files being corrupted either during creation or the upload of the tar archive. A new version of the dataset with the input files has been uploaded and tested for correctnes.

Furthermore, to facilitate running the test case, a shell script has been made made available with WaterSip that will download the test case, build the file structure implied in the INPUT file, compile watersip, and run the analysis. This should allow users to more easily run the test case.

Finally, I have changed the time range of the test case by 5 days, since the previous period started in the middle of a large precipitation event related to a midlatitude cyclone in the domain. The new test case period from 5-15 August 2022 has been used in all figures displaying the test case.

*2) A new version of FLEXPART has been released recently (FLEXPART11, Bakels et al., 2024). Does WaterSip work with this new version as well, or only the 10.4 version? Since the new version writes trajectory output to NetCDF files, it would probably require rewriting the routines reading the input data. However, it might be worth it because FLEXPART11 has several advantages compared to FLEXPART10.4, for example (relevant for Waterip) improved trajectory accuracy and the option to write out average instead of instantaneous values along trajectories.*

**Reply:** This is an interesting question. Currently, WaterSip is not able to read NetCDF files from FLEXPART11. The simplest way to make this work is to create a post-processor that will convert netCDF FLEXPART11 particle files to the binary dump format used with FLEXPART10.4 and before. This information has been added to Section 3.1 in the revised manuscript.

*3) The section on errors and uncertainties is very short, but I think it is an important section and should be extended a bit. I would suggest to add a few figures showing the sensitivity of the results for the Scandinavia case to the settings, specifically the timeStep, the uptakeThreshold and precipThreshold, and arrivalRHMin. This would be very useful for new users to understand the influence of these settings. Also some potential error sources are currently not mentioned in this section, e.g. the fact that WaterSip always assumes either only moisture uptakes or only moisture losses during one timestep but not both. This could lead to an overestimation of remote and an underestimation of local moisture sources. Or, when using WaterSip as a diagnostic for surface evaporation, the assumption that water evaporated from the location (lat,lon) where the moisture is taken up by the air parcel might not always be true.*

**Reply:** The additional error sources mentioned by the reviewer are now discussed in the revised manuscript in Sec. 6.1. Regarding the suggested sensitivity analysis, a similar comment has been made by the other reviewer. While the general approach has been to refer to systematic sensitivity tests done in earlier studies, a summary of several sensitivity tests are now presented in Sec. 6 for the parameters timeStep, trajLen, uptakeThreshold/precip Threshold, and arrivalRHMin in terms of arrival domain averages of the Lagrangian precipitation estimate, moisture source distance, and total accounted fraction.

*4) I did not fully understand the (difference between the) moisture source and transport quantities. For the moisture source quantities, the values are multiplied by Δq * f * m, for the transport quantities, they are multiplied only by f * m. I think the problem is that I don't really know what f and Δq are in this case. Are these the values after the accounting? If so, isn't f exactly Δq/q_0? So why is then f multiplied by Δq again? I am sure this is all done correctly in the code, but it could be explained a bit better (see also my specific comments on Equations 19 & 20).*

**Reply:** The difference between source and transport quantities is that the weighting of the source quantities is only done for each source region, whereas the transport quantities are also gridded at each time step during the airmass transport. The weights for the different quantities has been changed from the previous manuscript and software version, and is now correctly stated in the manuscript. The output fields "particle mass density"

and "moisture transport" are now described with these updated units. For the forward projection, the wrong weights were given in the previous version and are now corrected in Eq. 19 and 20.

**Specific comments**

*L9-11: This is a repetition of L2-4*

**Reply:** This sentence was removed in the revised manuscript

*L22: Bracket around Stohl and James (2004)*

**Reply:** Corrected

*L74: chose -> choose*

**Reply:** Corrected
*L108: Figure reference broken*

**Reply:** Corrected

*L133: doe -> does*

**Reply:** Corrected

*L151: What do you mean by „in case trajectories are used rather than air parcels"?*

**Reply:** Rephrased to "in case trajectory model output is used rather than particle dispersion model output"

*L154: I think this first sentence is not needed here.*

**Reply:** The sentence was removed and an absolute value notation was added to Eq. 10 and 11.

*Equation 12: Shouldn't this be the sum over i?*

**Reply:** Yes, this has been corrected in the revised manuscript. Also, M is now introduced in the context of Eq. 12.

*L168: This is interesting. Why does this (f_tot > 1) happen? This would be a case where the algorithm from Dütsch et al. (2018) is not equivalent, because there f_tot is by definition always <= 1.*

**Reply:** Yes, the summation should be over i. The f_tot > 1 can happen due to rainout events that are below the precipThreshold value. This explanation has been added in the revised manuscript.

*L233: Why „differences in"? Not just the source contributions themselves?*

**Reply:** Yes, this has been rephrased for clarity.

*L266: Do you mean atmospheric properties? Because the positions come from the trajectory calculation, so that would be the second part of the sentence.*

**Reply:** Yes, corrected.

*L314: What would be a reason for setting a maximum relative humidity?*

**Reply:** A hypothetical example would be in the tracing of vapour rather than precipitation properties, where one could be interested in cloud-free arrival locations only. This has been added in the revision.

*L323: partcels -> parcels*

**Reply:** Corrected

*L360-369: I didn't understand this part with the gridRadius.*

**Reply:** The main point here is that the gridding is not done for the grid cell closest to the air parcel location, but distributed to surrounding grid cells using the areal overlap between the gridding radius and neighbouring grid cells. Thus, the gridding radius serves the purpose to distribute the information on the neighboring grid cells according to distance/position. This section has been rephrased for clarity.

*L395: Could you briefly explain what these methods do, specifically Gustafsson et al. (2010) and Dirmeyer and Brubaker (1999)?*

**Reply:** The concept of the methods by Gustafsson et al. (2010) and Dirmeyer and Brubaker (1999) are now briefly explained in the revised manuscript.

*L403: Remove „are"*

**Reply:** Corrected

*L436: Closing bracket missing*

**Reply:** Corrected

*L467: chose -> choose*

**Reply:** Corrected

*L483: Maybe start with a brief description of the meteorological situation for the event?*

**Reply:** A short description of the meteorological situation during the event has been added.

*L498 (and others): Sometimes day is d, dy, or day. Please be consistent.*

**Reply:** Now d is used as units throughout.

*L515: Stohl et al. (2005) or Stohl and James (2004)?*

**Reply:** The correct reference here is Stohl and James (2005).

*L539-L545: I don't understand how the quantities moisture transport, air mass mixing, and rainout are obtained. Could you explain this better? For example, what is meant by gridded product of the specific humidity?*

**Reply:** The explanation of transport quantities has been rephrased and extended by explicitly including the weight during gridding for each quantity in the revised manuscript.

*L551: I don't understand this first sentence. Do you mean differences _in_ moisture sources and transport? But moisture sources are not shown here… (?)*

**Reply:** The intention was to talk about moisture source differences within the arrival domain. Such differences can indeed not be made visible by arrival quantities, but by LFP quantities introduced below. This has been rephrased for clarify.

*L554: quantity -> quantities*

**Reply:** Corrected

*L554-L557: How does the precipitation estimate by WaterSip compare to ERA5 precipitation? This would be a good validity check.*

**Reply:** See reply to major comment #3

*Equation 19: Shouldn't the denominator be P_tot (without k)? And the enumerator would correspond to T^k_0 * P^k_tot?*

**Reply:** Yes, that is correct, thank you for pointing that out. The enumerator has also been revised as suggested since this formulation is easier to understand.

*L579: Earlier λ and φ were used for lon and lat. Please use consistent notation throughout the manuscript. How is the mean over longitudes calculated? Are the coordinates converted to a Cartesian grid first? Otherwise there would be problems for e.g. lon1=-179 and lon2=179.*

**Reply:** Yes, the longitude averaging is done using conversion to a Cartesian grid. The notation will be adjusted to be consistent throughout as suggested.

*Equation 20: What is M? I assume the time steps of the trajectories? You could use this also in Equation 12 for consistency. Shouldn't Δq also appear in the denominator?*

**Reply:** Actually no, M and N are different and that should have been stated here. Index M has been changed to U and runs only over all uptake events, whereas N runs over all trajectory time steps. This has now been added to the text describing Eq. 20, and U is introduced at Eq. 12.

*L583: zeta -> \zeta*

**Reply:** Corrected

*L595: This is a detail, but wouldn't it be better to use the land fraction itself in the averaging (instead of 0 or 1)?*

**Reply:** Interesting comment. Conceptually, WaterSip now defines grid points as either land or sea, based on the land-sea fraction. It could be possible to use land fraction directly to calculate the average. Probably the differences to the current approach will be miniscule, but it may be worth considering that for a future update.

*L600: and Eq. 19*

**Reply:** Weighted moisture source temperatures are obtained using Eq. 20.

*L604 and L617: „the" too much.*

**Reply:** Corrected

*L607: Maybe briefly explain what d-excess is or define it?*

**Reply:** The use of the d-excess is now briefly explained in the revised manuscript.

*Equation 22: Is N the same as M in Equation 20? If so, use M instead?*

**Reply:** Actually no, M and N are different and that should have been stated with Eq. 20. Index M has been changed to U and runs only over all uptake events, whereas N runs over all trajectory time steps. This has now been added to the text describing Eq. 20, and U is introduced with Eq. 12.

*L650: where -> were*

**Reply:** Corrected

*L698: Would it be possible to provide a script for reading the .traj files? Otherwise this is difficult because of the binary format.*

**Reply:** Yes, a jupyter notebook has been added to the WaterSip repository that allows to read in traj format files, for convertion to an xarray dataset and plotting. This is now mentioned in the manuscript.

*L763: arrival _in the_ target region?*

**Reply:** Corrected

*L780: The link to the WaterSip source code does not work. The one given on Zenodo is correct.*

**Reply:** The link has been corrected to https://git.app.uib.no/GFI-public/watersip.

**Figure and table comments**

*Figures 5,6,7: I would use a different colormap. This one looks like topography and is not very intuitive for sequential values (e.g. white means high).*

**Reply:** The figures have been redrawn with colormaps that provide sequential interpretation. The accounted fraction is drawn with a categorical colormap to facilitate interpretation of the percentage explained by the different fractions.

*Figure 9: The locations of the colors don't correspond to the labels of the colorbar. Which one is correct?*

**Reply:** The labels were flipped in the figure, and the coloring was not assigned correctly. The figure has now been redrawn with correct colors and matching labels.

*Figure 10: Are these quantities weighted by f? From Figure 10 it seems so, but the text doesn't mention it. In (d) where can we see the time before arrival mentioned in the caption?*

**Reply:** Yes, but the weight differs by quantity. The respective weights have been added to Table 3 and the text has been clarified. The caption to Fig. 10(d) now says that it gives the PDF as a function of distance.

*Table 3: Why do the uptakes have negative values?*

**Reply:** The negative quantity is actually time before arrival in hours, rather than uptake, which is the gridded quantity. The table has been corrected by using the title "uptake time" and so on, instead of uptake.

*Figure 11b: I am not sure what is shown here. The figure says specific humidity, but the text says total accounted fraction. Where do we see that „moisture uptakes of more than 10 days before arrival are mostly overwritten by later uptake events"? I would suggest to use a different colormap, because red for moist and blue for wet is a bit counterintuitive. Also jet has many other problems (irregular lightness, not perceptually uniform, not colorblind-friendly).*

**Reply:** The figure gives the correct information, the text has been corrected and clarified in the revised manuscript. The figure has been redrawn with more colorblind-friendly colormaps.

**References**

*Bakels, L., Tatsii, D., Tipka, A., Thompson, R., Dütsch, M., Blaschek, M., Seibert, P., Baier, K., Bucci, S., Cassiani, M., Eckhardt, S., Groot Zwaaftink, C., Henne, S., Kaufmann, P., Lechner, V., Maurer, C., Mulder, M. D., Pisso, I., Plach, A., Subramanian, R., Vojta, M., and Stohl, A.: FLEXPART version 11: improved accuracy, efficiency, and flexibility, Geoscientific Model Development, 17, 7595–7627, https://doi.org/10.5194/gmd-17-7595-2024, 2024.*

*Dütsch, M., Pfahl, S., Meyer, M., and Wernli, H.: Lagrangian process attribution of isotopic variations in near-surface water vapour in a 30-year regional climate simulation over Europe, Atmospheric Chemistry and Physics, 18, 1653–1669, https://doi.org/10.5194/acp-18-1653-2018, 2018.*

*Sodemann, H., Schwierz, C., and Wernli, H.: Interannual variability of Greenland winter precipitation sources: Lagrangian moisture diagnostic and North Atlantic Oscillation influence, J. Geophys. Res., 113, D03 107, http://dx.doi.org/10.1029/2007JD008503, 2008.*